# EFFICIENT ACTION ROBUST REINFORCEMENT LEARNING WITH PROBABILISTIC POLICY EXECUTION UNCERTAINTY

## ABSTRACT

Robust reinforcement learning (RL) aims to find a policy that optimizes the worst-case performance in the face of uncertainties. In this paper, we focus on action robust RL with the probabilistic policy execution uncertainty, in which, instead of always carrying out the action specified by the policy, the agent will take the action specified by the policy with probability $1 - \rho$ and an alternative adversarial action with probability $\rho$. We establish the existence of an optimal policy on the action robust MDPs with probabilistic policy execution uncertainty and provide the action robust Bellman optimality equation for its solution. Furthermore, we develop Action Robust Reinforcement Learning with Certificates (ARRLC) algorithm that achieves minimax optimal regret and sample complexity. Furthermore, we conduct numerical experiments to validate our approach's robustness, demonstrating that ARRLC outperforms non-robust RL algorithms and converges faster than the robust TD algorithm in the presence of action perturbations.

## 1 INTRODUCTION

Reinforcement learning (RL), a framework of control-theoretic problem that makes decisions over time under an unknown environment, has many applications in a variety of scenarios such as recommendation systems (Zhao et al., 2018), autonomous driving (O' Kelly et al., 2018), finance (Liu et al., 2020) and business management (Nazari et al., 2018), to name a few. However, the solutions to standard RL methods are not inherently robust to uncertainties, perturbations, or structural changes in the environment, which are frequently observed in real-world settings. A trustworthy reinforcement learning algorithm should be competent in solving challenging real-world problems with robustness against perturbations and uncertainties. Robust RL aims to improve the worst-case performance of algorithms deterministically or statistically in the face of uncertainties in different MDP components, including observations/states (Zhang et al., 2020a; Sun et al., 2022), actions (Tessler et al., 2019; Klima et al., 2019), transitions (Nilim & El Ghaoui, 2005; Iyengar, 2005; Tamar et al., 2014; Wang & Zou, 2021), and rewards (Huang & Zhu, 2019; Lecarpentier & Rachelson, 2019).

In this paper, we consider action uncertainties, also called policy execution uncertainties, and probabilistic uncertainty set proposed in (Tessler et al., 2019). Robust RL against action uncertainties focuses on the discrepancy between the actions generated by the RL agent and the conducted actions. Taking the robot control as an example, such policy execution uncertainty may come from the actuator noise, limited power range, or actuator failures in the real world. Taking the medication advice in healthcare as another example, such policy execution uncertainty may come from the patient's personal behaviors like drug refusal, forgotten medication, or overdose etc.

To deal with the policy execution uncertainty, robust RL methods (Pinto et al., 2017; Tessler et al., 2019) adopt the adversarial training framework (Goodfellow et al., 2014; Madry et al., 2018) and assume an adversary conducting adversarial attacks to mimic the naturalistic uncertainties. Training with an adversary can naturally be formulated as a zero-sum game between the adversary and the RL agent. However, these interesting works do not provide theoretical guarantee on sample complexity or regret. In this paper, we aim to fill this gap. The approaches in (Pinto et al., 2017; Tessler et al., 2019) iteratively apply two stages: (i) given a fixed adversary policy, it calculates the agent's optimal policy; and (ii) update the adversary policy against the updated agent's policy. The repetition of stage

(i) requires repeatedly solving MDP to find the optimal policy, which is sample inefficient. Motivated by the recent theoretical works on transition probability uncertainty that use the robust dynamic programming method Iyengar (2005) and achieve efficient sample complexity Wang & Zou (2021); Panaganti & Kalathil (2022); Xu et al. (2023), we introduce the action robust Bellman equations and design sample efficient algorithms based on the action robust Bellman equations. Our methods simultaneously update the adversary policy and agent's policy instead of updating one after another is converged. Our major contributions are summarized as follows:

- We show that the robust problem can be solved by the iteration of the action robust Bellman optimality equations. Motivated by this, we design two efficient algorithms.
- We develop a model-based algorithm, Action Robust Reinforcement Learning with Certificates (ARRLC), for episodic action robust MDPs, and show that it achieves minimax order optimal regret and minimax order optimal sample complexity.
- We develop a model-free algorithm for episodic action robust MDPs, and analyze its regret and sample complexity. Due to space limitations, this result is shown in the Appendix D.
- We conduct numerical experiments to validate the robustness of our approach. In our experiments, our robust algorithm achieves a much higher reward than the non-robust RL algorithm when being tested with some action perturbations; and our ARRLC algorithm converges much faster than the robust TD algorithm in (Klima et al., 2019).

## 2 RELATED WORK

We mostly focus on papers that are related to sample complexity bounds for the episodic RL and the two-player zero-sum Markov game, and action robust RL, that are close related to our model. We remark that there are also related settings, e.g., infinite-horizon discounted MDP (Li et al., 2020; He et al., 2021), robust RL with other uncertainties (Iyengar, 2005; Lecarpentier & Rachelson, 2019; Zhang et al., 2020a; Wang & Zou, 2021), robust offline RL (Guo et al., 2022; Shi & Chi, 2022), adversarial training with a generative RL model (Xu et al., 2023; Panaganti & Kalathil, 2022), adversarial attacks on RL (Zhang et al., 2020b; Liu & Lai, 2021; Sun et al., 2021), etc. These settings are beyond the scope of this paper, though our techniques may be also related to these settings.

**Action robust RL.** (Pinto et al., 2017) introduce robust adversarial reinforcement learning to address the generalization issues in reinforcement learning by training with a destabilizing adversary that applies disturbance forces to the system. (Tessler et al., 2019) introduce two new criteria of robustness for reinforcement learning in the face of action uncertainty. We follow its probabilistic action robust MDP (PR-MDP) in which, instead of the action specified by the policy, an alternative adversarial action is taken with probability $\rho$. They generalize their policy iteration approach to deep reinforcement learning (DRL) and provide extensive experiments. A similar uncertainty setting was presented (Klima et al., 2019), which extends temporal difference (TD) learning algorithms by a new robust operator and shows that the new algorithms converge to the optimal robust $Q$-function. However, no theoretical guarantee on sample complexity or regret is provided in these works. We develop a minimax sample efficient algorithm and fill this gap.

**Sample complexity bounds for the episodic RL.** There is a rich literature on sample complexity guarantees for episodic tabular RL, for example (Kearns & Singh, 2002; Strehl et al., 2006; Auer et al., 2008; Azar et al., 2017; Dann et al., 2017; Jin et al., 2018; Dann et al., 2019; Simchowitz & Jamieson, 2019; Zhang et al., 2020c; 2021). However, these methods can not be directly applied in action robust MDP with small technical changes. Most relevant to our paper is the work about policy certificates (Dann et al., 2019). The algorithm ORLC in (Dann et al., 2019) calculate both the upper bound and lower bound of the value functions, and outputs policy certificates that bound the sub-optimality and return of the policy. Our proposed ARRLC shares a similar structure with ORLC, but we develop new adversarial trajectory sampling and action robust value iteration method in ARRLC, and new techniques to bound the sum of variances so that our algorithm suits for action robust MDPs.

**Sample complexity bounds for the two-player zero-sum Markov game.** Training with an adversary can naturally be formulated as a zero-sum game between the adversary and the RL agent. Some sample efficient algorithms for two-player zero-sum Markov game can be used to train the action robust RL agent. The efficient multi-agent RL algorithms, like (Liu et al., 2021; Jin et al., 2021), can

be used to solve the action robust optimal policy but are not minimax optimal. They are a factor of $A$ or $H^2$ above the minimax lower bound. Our algorithm ARRLC is minimax optimal.

## 3    PROBLEM FORMULATION

**Tabular MDPs.** We consider a tabular episodic MDP $\mathcal{M} = (\mathcal{S}, \mathcal{A}, H, P, R)$, where $\mathcal{S}$ is the state space with $|\mathcal{S}| = S$, $\mathcal{A}$ is the action space with $|\mathcal{A}| = A$, $H \in \mathbb{Z}^+$ is the number of steps in each episode, $P$ is the transition matrix so that $P_h(\cdot|s, a)$ represents the probability distribution over states if action $a$ is taken for state $s$ at step $h \in [H]$, and $R_h : \mathcal{S} \times \mathcal{A} \to [0, 1]$ represents the reward function at the step $h$. In this paper, the probability transition functions and the reward functions can be different at different steps.

The agent interacts with the MDP in episodes indexed by $k$. Each episode $k$ is a trajectory $\{s_1^k, a_1^k, r_1^k, \cdots, s_H^k, a_H^k, r_H^k\}$ of $H$ states $s_h^k \in \mathcal{S}$, actions $a_h^k \in \mathcal{A}$, and rewards $r_h^k \in [0, 1]$. At each step $h \in [H]$ of episode $k$, the agent observes the state $s_h^k$ and chooses an action $a_h^k$. After receiving the action, the environment generates a random reward $r_h^k \in [0, 1]$ derived from a distribution with mean $R_h(s_h^k, a_h^k)$ and next state $s_{h+1}^k$ that is drawn from the distribution $P_h(\cdot|s_h^k, a_h^k)$. For notational simplicity, we assume that the initial states $s_1^k = s_1$ are deterministic in different episode $k$.

A (stochastic) Markov policy of the agent is a set of $H$ maps $\pi := \{\pi_h : \mathcal{S} \to \Delta_{\mathcal{A}}\}_{h \in [H]}$, where $\Delta_{\mathcal{A}}$ denotes the simplex over $\mathcal{A}$. We use notation $\pi_h(a|s)$ to denote the probability of taking action $a$ in state $s$ under stochastic policy $\pi$ at step $h$. A deterministic policy is a policy that maps each state to a particular action. Therefore, when it is clear from the context, we abuse the notation $\pi_h(s)$ for a deterministic policy $\pi$ to denote the action $a$ which satisfies $\pi_h(a|s) = 1$.

**Action robust MDPs.** In the action robust case, the policy execution is not accurate and lies in some uncertainty set centered on the agent's policy $\pi$. Denote the actual behavior policy by $\widetilde{\pi}$ where $\widetilde{\pi} \in \Pi(\pi)$ and $\Pi(\pi)$ is the uncertainty set of the policy execution. Denote the actual behavior action at episode $k$ and step $h$ by $\widetilde{a}_h^k$ where $\widetilde{a}_h^k \sim \widetilde{\pi}_h^k$. Define the action robust value function of a policy $\pi$ as the worst-case expected accumulated reward over following any policy in the uncertainty set $\Pi(\pi)$ centered on a fixed policy $\pi$:

$$V_h^\pi(s) = \min_{\widetilde{\pi} \in \Pi(\pi)} \mathbb{E}\left[\sum_{h'=h}^H R_{h'}(s_{h'}, a_{h'})|s_h = s, a_{h'} \sim \widetilde{\pi}_{h'}(\cdot|s_{h'}), \forall h' > h\right]. \tag{1}$$

$V_h^\pi$ represents the action robust value function of policy $\pi$ at step $h$. Similarly, define the action robust $Q$-function of a policy $\pi$:

$$Q_h^\pi(s, a) = \min_{\widetilde{\pi} \in \Pi(\pi)} \mathbb{E}\left[\sum_{h'=h}^H R_{h'}(s_{h'}, a_{h'})|s_h = s, a_h = a, a_{h'} \sim \widetilde{\pi}_{h'}(\cdot|s_{h'}), \forall h' > h\right]. \tag{2}$$

The goal of action robust RL is to find the optimal robust policy $\pi^*$ that maximizes the worst-case accumulated reward: $\pi^* = \arg\max_\pi V_1^\pi(s), \forall s \in \mathcal{S}$. We also denote $V^{\pi^*}$ and $Q^{\pi^*}$ by $V^*$ and $Q^*$.

**Probabilistic policy execution uncertain set.** We follow the setting of the probabilistic action robust MDP (PR-MDP) introduced in (Tessler et al., 2019) to construct the probabilistic policy execution uncertain set. For some $0 \leq \rho \leq 1$, the policy execution uncertain set is defined as:

$$\Pi^\rho(\pi) := \{\widetilde{\pi} : \forall s, \forall h, \exists \pi_h'(\cdot|s) \in \Delta_{\mathcal{A}} \text{ such that } \widetilde{\pi}_h(\cdot|s) = (1 - \rho)\pi_h(\cdot|s) + \rho\pi_h'(\cdot|s)\}. \tag{3}$$

The policy execution uncertain set can be even simpler expressed as $\Pi^\rho(\pi) = (1 - \rho)\pi + \rho(\Delta_{\mathcal{A}})^{S \times H}$.

In this setting, an optimal probabilistic robust policy is optimal w.r.t. a scenario in which, with probability at most $\rho$, an adversary takes control and performs the worst possible action. We call $\pi'$ as the adversarial policy. For different agent's policy $\pi$, the corresponding adversarial policy $\pi'$ that minimizes the cumulative reward may be different.

**Additional notations.** We set $\iota = \log(2SAHK/\delta)$ for $\delta > 0$. For simplicity of notation, we treat $P$ as a linear operator such that $[P_h V](s, a) := \mathbb{E}_{s' \sim P_h(\cdot|s, a)} V(s')$, and we define two additional operators $\mathbb{D}$ and $\mathbb{V}$ as follows: $[\mathbb{D}_{\pi_h} Q](s) := \mathbb{E}_{a \sim \pi_h(\cdot|s)} Q(s, a)$ and $\mathbb{V}_{P_h} V_{h+1}(s, a) := \sum_{s'} P_h(s'|s, a)\left(V_{h+1}(s') - [P_h V_{h+1}](s, a)\right)^2 = [P_h(V_{h+1})^2](s, a) - ([P_h V_{h+1}](s, a))^2$.

## 4 EXISTENCE OF THE OPTIMAL ROBUST POLICY

For the standard tabular MDPs, when the state space, action space, and the horizon are all finite, there always exists an optimal policy. In addition, if the reward functions and the transition probabilities are known to the agent, the optimal policy can be solved by solving the Bellman optimality equation. In the following theorem, we show that the optimal policy also always exists in action robust MDPs and can be solved by the action robust Bellman optimality equation.

**Proposition 1** *If the uncertainty set of the policy execution has the form in* (3)*, the following perfect duality holds for all $s \in \mathcal{S}$ and all $h \in [H]$:*

$$\max_{\pi} \min_{\widetilde{\pi} \in \Pi^{\rho}(\pi)} \mathbb{E}\left[\sum_{h'=h}^{H} R_{h'}(s_{h'}, a_{h'})|s_h = s, a_{h'} \sim \widetilde{\pi}_{h'}(\cdot|s_{h'})\right]$$

$$= \min_{\widetilde{\pi} \in \Pi^{\rho}(\pi)} \max_{\pi} \mathbb{E}\left[\sum_{h'=h}^{H} R_{h'}(s_{h'}, a_{h'})|s_h = s, a_{h'} \sim \widetilde{\pi}_{h'}(\cdot|s_{h'})\right]. \quad (4)$$

*There always exists a deterministic optimal robust policy $\pi^*$. The problem can be solved by the iteration of the action robust Bellman optimality equation on $h = H, \cdots, 1$. The action robust Bellman equation and the action robust Bellman optimality equation are:*

$$\begin{cases} V_h^{\pi}(s) = (1-\rho)[\mathbb{D}_{\pi_h} Q_h^{\pi}](s) + \rho \min_{a \in \mathcal{A}} Q_h^{\pi}(s, a) \\ Q_h^{\pi}(s, a) = R_h(s, a) + [P_h V_{h+1}^{\pi}](s, a) \\ V_{H+1}^{\pi}(s) = 0, \ \forall s \in \mathcal{S} \end{cases} \quad (5)$$

$$\begin{cases} V_h^*(s) = (1-\rho) \max_{a \in \mathcal{A}} Q_h^*(s, a) + \rho \min_{b \in \mathcal{A}} Q_h^*(s, b) \\ Q_h^*(s, a) = R_h(s, a) + [P_h V_{h+1}^*](s, a) \\ V_{H+1}^*(s) = 0, \ \forall s \in \mathcal{S} \end{cases} \quad (6)$$

We define $C_h^{\pi,\pi',\rho}(s) := \mathbb{E}\left[\sum_{h'=h}^{H} R_{h'}(s_{h'}, a_{h'})|s_h = s, a_{h'} \sim \widetilde{\pi}_{h'}(\cdot|s_{h'})\right]$. The perfect duality of the control problems in (4) is equivalent to $\max_{\pi} \min_{\pi'} C_h^{\pi,\pi',\rho}(s) = \min_{\pi'} \max_{\pi} C_h^{\pi,\pi',\rho}(s)$. We provide the detailed proof of the perfect duality and the existence of the optimal policy in Appendix B. Our proposed model-based algorithm in Section 5 and model-free algorithm in Appendix D are based on the action robust Bellman optimality equation. Using the iteration of the proposed action robust Bellman equation to solve the robust problem can simultaneously update the adversary policy and agent policy and avoid inefficient alternating updates.

## 5 ALGORITHM AND MAIN RESULTS

In this section, we introduce the proposed **A**ction **R**obust **R**einforcement **L**earning with **C**ertificates (ARRLC) algorithm and provides its theoretical guarantee. The pseudo code is listed in Algorithm 1. Here, we highlight the main idea of our algorithm. Algorithm 1 trains the agent in a clean (simulation) environment and learns a policy that performs well when applied to a perturbed environment with probabilistic policy execution uncertainty. To simulate the action perturbation, Algorithm 1 chooses an adversarial action with probability $\rho$. To learn the agent's optimal policy and the corresponding adversarial policy, Algorithm 1 computes an optimistic estimate $\overline{Q}$ of $Q^*$ and a pessimistic estimate $\underline{Q}$ of $Q^{\overline{\pi}^k}$. Algorithm 1 uses the optimistic estimates to explore the possible optimal policy $\overline{\pi}$ and uses the pessimistic estimates to explore the possible adversarial policy $\underline{\pi}$. As shown later in Lemma 2, $\overline{V} \geq V^* \geq V^{\overline{\pi}} \geq \underline{V}$ holds with high probabilities. The optimistic and pessimistic estimates $\overline{V}$ and $\underline{V}$ can provide policy certificates, which bounds the cumulative rewards of the return policy $\overline{\pi}^k$ and $\overline{V} - \underline{V}$ bounds the sub-optimality of the return policy $\overline{\pi}^k$ with high probabilities. The policy certificates can give us some insights about the performance of $\overline{\pi}^k$ in the perturbed environment with probabilistic policy execution uncertainty.

**Algorithm 1:** ARRLC (**A**ction **R**obust **R**einforcement **L**earning with **C**ertificates)

1: Initialize $\overline{V}_h(s) = H - h + 1, \overline{Q}_h(s,a) = H - h + 1, \underline{V}_h(s) = 0, \underline{Q}_h(s,a) = 0, \hat{r}_h(s,a),$
   $N_h(s,a) = 0$ and $N_h(s,a,s') = 0$ for any state $s \in \mathcal{S}$, any action $a \in \mathcal{A}$ and any step $h \in [H]$.
   $\overline{V}_{H+1}(s) = \underline{V}_{H+1}(s) = 0$ and $\overline{Q}_{H+1}(s,a) = \underline{Q}_{H+1}(s,a) = 0$ for any $s$ and $a$. $\Delta = H$.
2: **for** episode $k = 1, 2, \ldots, K$ **do**
3:    **for** step $h = 1, 2, \ldots, H$ **do**
4:       Observe $s_h^k$.
5:       Set $\overline{\pi}_h^k(s) = \arg\max_a \overline{Q}_h(s,a)$ , $\underline{\pi}_h^k(s) = \arg\min_a \underline{Q}_h(s,a)$, $\widetilde{\pi}_h^k = (1-\rho)\overline{\pi}_h^k + \rho\underline{\pi}_h^k$.
6:       Take action $a_h^k \sim \widetilde{\pi}_h^k(\cdot|s_h^k)$.
7:       Receive reward $r_h^k$ and observe $s_{h+1}^k$.
8:       Set $N_h(s_h^k, a_h^k) \leftarrow N_h(s_h^k, a_h^k) + 1$, $N_h(s_h^k, a_h^k, s_{h+1}^k) \leftarrow N_h(s_h^k, a_h^k, s_{h+1}^k) + 1$.
9:       Set $\hat{r}_h^k(s_h^k, a_h^k) \leftarrow \hat{r}_h^k(s_h^k, a_h^k) + (r_h^k - \hat{r}_h^k(s_h^k, a_h^k))/N_h(s_h^k, a_h^k)$.
10:      Set $\hat{P}_h(\cdot|s_h^k, a_h^k) = N_h(s_h^k, a_h^k, \cdot)/N_h(s_h^k, a_h^k)$.
11:    **end for**
12:    **Output** policy $\overline{\pi}^k$ with certificates $\mathcal{I}_k = [\underline{V}_1(s_1^k), \overline{V}_1(s_1^k)]$ and $\epsilon_k = |\mathcal{I}_k|$.
13:    **if** $\epsilon_k < \Delta$ **then**
14:       $\Delta \leftarrow \epsilon_k$ and $\pi^{out} \leftarrow \overline{\pi}^k$.
15:    **end if**
16:    **for** step $h = H, H-1, \ldots, 1$ **do**
17:       **for** each $(s,a) \in \mathcal{S} \times \mathcal{A}$ with $N_h(s,a) > 0$ **do**
18:          Set $\theta_h(s,a) =$

$$\sqrt{\frac{2\mathbb{V}_{\hat{P}_h}[(\overline{V}_{h+1}+\underline{V}_{h+1})/2](s,a)\iota}{N_h(s,a)}} + \sqrt{\frac{2\hat{r}_h(s,a)\iota}{N_h(s,a)}} + \frac{\hat{P}_h(\overline{V}_{h+1}-\underline{V}_{h+1})(s,a)}{H} + \frac{(24H^2+7H+7)\iota}{3N_h(s,a)},$$

19:          $\overline{Q}_h(s,a) \leftarrow \min\{H-h+1, \hat{r}_h(s,a) + \hat{P}_h\overline{V}_{h+1}(s,a) + \theta_h(s,a)\}$,
20:          $\underline{Q}_h(s,a) \leftarrow \max\{0, \hat{r}_h(s,a) + \hat{P}_h\underline{V}_{h+1}(s,a) - \theta_h(s,a)\}$,
21:          $\overline{\pi}_h^{k+1}(s) = \arg\max_a \overline{Q}_h(s,a)$ , $\underline{\pi}_h^{k+1}(s) = \arg\min_a \underline{Q}_h(s,a)$,
22:          $\overline{V}_h(s) \leftarrow (1-\rho)\overline{Q}_h(s, \overline{\pi}_h^{k+1}(s)) + \rho\overline{Q}_h(s, \underline{\pi}_h^{k+1}(s))$,
23:          $\underline{V}_h(s) \leftarrow (1-\rho)\underline{Q}_h(s, \overline{\pi}_h^{k+1}(s)) + \rho\underline{Q}_h(s, \underline{\pi}_h^{k+1}(s))$.
24:       **end for**
25:    **end for**
26: **end for**
27: **return** $\pi^{out}$

## 5.1 ALGORITHM DESCRIPTION

We now describe the proposed ARRLC algorithm in more details. In each episode, the ARRLC algorithm can be decomposed into two parts.

- Line 3-11 (Sample trajectory and update the model estimate): Simulates the action robust MDP, executes the behavior policy $\widetilde{\pi}$, collects samples, and updates the estimate of the reward and the transition.

- Line 16-25 (Adversarial planning from the estimated model): Performs value iteration with bonus to estimate the robust value functions using the empirical estimate of the transition $\hat{P}$, computes a new policy $\overline{\pi}$ that is optimal respect to the estimated robust value functions, and computes a new optimal adversarial policy $\underline{\pi}$ respect to the agent's policy $\overline{\pi}$.

At a high-level, this two-phase policy is standard in the majority of model-based RL algorithms (Azar et al., 2017; Dann et al., 2019). Algorithm 1 shares similar structure with ORLC (Optimistic Reinforcement Learning with Certificates) in (Dann et al., 2019) but has some significant differences in line 5-6 and line 18-23. The first main difference is that the ARRLC algorithm simulates the probabilistic policy execution uncertainty by choosing an adversarial action with probability $\rho$. The adversarial policy and the adversarial actions are computed by the ARRLC algorithm. The second main difference is that the ARRLC algorithm simultaneously plans the agent policy $\overline{\pi}$ and the adversarial policy $\underline{\pi}$ by the action robust Bellman optimality equation.

These two main difference brings two main challenges in the design and analysis of our algorithm.

(1) The ARRLC algorithm simultaneously plans the agent policy and the adversarial policy. However the planned adversarial policy $\underline{\pi}$ is not necessarily the true optimal adversary policy towards the agent policy $\overline{\pi}$ because of the estimation error of the value functions. We carefully design the bonus items and the update role of the value functions so that $\overline{V}_h(s) \geq V_h^*(s) \geq V_h^{\overline{\pi}}(s) \geq \underline{V}_h(s)$ and $\overline{Q}_h(s,a) \geq Q_h^*(s,a) \geq Q_h^{\overline{\pi}}(s,a) \geq \underline{Q}_h(s,a)$ hold for all $s$ and $a$.

(2) A crucial step in many UCB-type algorithms based on Bernstein inequality is bounding the sum of variance of estimated value function across the planning horizon. The behavior policies in these UCB-type algorithms are deterministic. However, the behavior policy in our ARRLC algorithm is not deterministic due to the simulation of the adversary's behavior. The total variance is the weighted sum of the sum of variance of estimated value function across two trajectories. Even if action $\overline{\pi}(s_h^k)$ or $\underline{\pi}(s_h^k)$ is not sampled at state $s_h^k$, it counts in the total variance. Thus, the sum of variance is no longer simply the variance of the sum of rewards per episode, and new techniques are introduced. For example, the variance of $\overline{V} + \underline{V}$ can be connected to the variance of $C^{\pi^{k*}, \underline{\pi}^k, \rho}$, where $\pi^{k*}$ is the optimal policy towards the adversary policy $\underline{\pi}^k$ with $\pi_h^{k*}(s) = \arg\max_\pi C_h^{\pi, \underline{\pi}^k, \rho}(s)$. Then the variance of $C^{\pi^{k*}, \underline{\pi}^k, \rho}$ can be bounded via recursion on the sampled trajectories.

## 5.2 THEORETICAL GUARANTEE

We define the cumulative regret of the output policy $\overline{\pi}^k$ at each episodes $k$ as $Regret(K) := \sum_{k=1}^{K}(V_1^*(s_1^k) - V_1^{\overline{\pi}^k}(s_1^k))$.

**Theorem 1** *For any $\delta \in (0,1]$, letting $\iota = \log(2SAHK/\delta)$, then with probability at least $1 - \delta$, Algorithm 1 achieves:*

- $V_1^*(s_1) - V_1^{\pi^{out}}(s_1) \leq \epsilon$, *if the number of episodes* $K \geq \Omega(SAH^3\iota^2/\epsilon^2 + S^2AH^3\iota^2/\epsilon)$.

- $Regret(K) = \sum_{k=1}^{K}(V_1^*(s_1^k) - V_1^{\overline{\pi}^k}(s_1^k)) \leq \mathcal{O}(\sqrt{SAH^3K}\iota + S^2AH^3\iota^2)$.

For small $\epsilon \leq H/S$, the sample complexity scales as $\mathcal{O}(SAH^3\iota^2/\epsilon^2)$. For the case with a large number of episodes $K \geq S^3AH^3\iota$, the regret scales as $\mathcal{O}(\sqrt{SAH^3K}\iota)$. For the standard MDPs, the information-theoretic sample complexity lower bound is $\Omega(SAH^3/\epsilon^2)$ provided in (Zhang et al., 2020c) and the regret lower bound is $\Omega(\sqrt{SAH^3K})$ provided in (Jin et al., 2018). When $\rho = 0$, the action robust MDPs is equivalent to the standard MDPs. Thus, the information-theoretic sample complexity lower bound and the regret lower bound of the action robust MDPs should have same dependency on $S$, $A$, $H$, $K$ or $\epsilon$. The lower bounds show the optimality of our algorithm up to logarithmic factors.

## 6 PROOF SKETCH

In this section, we provide sketch of the proof, which will highlight our the main ideas of our proof. First, we will show that $\overline{V}_h(s) \geq V_h^*(s) \geq V_h^{\overline{\pi}}(s) \geq \underline{V}_h(s)$ hold for all $s$ and $a$. Then, the regret can be bounded by $\overline{V}_1 - \underline{V}_1$ and then be divided by four items, each of which can then be bounded separately. The full proof can be found in the appendix contained in the supplementary material.

We first introduce a few notations. We use $\overline{Q}_h^k, \overline{V}_h^k, \underline{Q}_h^k, \underline{V}_h^k, N_h^k, \hat{P}_h^k, \hat{r}_h^k$ and $\theta_h^k$ to denote the values of $\overline{Q}_h, \overline{V}_h, \underline{Q}_h, \underline{V}_h, \max\{N_h, 1\}, \hat{P}_h, r_h$ and $\theta_h$ in the beginning of the $k$-th episode in Algorithm 1.

## 6.1 PROOF OF MONOTONICITY

We define $\mathcal{E}^R$ to be the event where

$$\left|\hat{r}_h^k(s,a) - R_h(s,a)\right| \leq \sqrt{\frac{2\hat{r}_h^k(s,a)\iota}{N_h^k(s,a)}} + \frac{7\iota}{3(N_h^k(s,a))} \tag{7}$$

holds for all $(s, a, h, k) \in S \times A \times [H] \times [K]$. We also define $\mathcal{E}^{PV}$ to be the event where

$$\left| (\hat{P}_h^k - P_h) V_{h+1}^*(s, a) \right| \leq \sqrt{\frac{2 \mathbb{V}_{\hat{P}_h^k} V_{h+1}^*(s, a) \iota}{N_h^k(s, a)}} + \frac{7H\iota}{3(N_h^k(s, a))} \tag{8}$$

$$\left| (\hat{P}_h^k - P_h) V_{h+1}^{\overline{\pi}^k}(s, a) \right| \leq \sqrt{\frac{2 \mathbb{V}_{\hat{P}_h^k} V_{h+1}^{\overline{\pi}^k}(s, a) \iota}{N_h^k(s, a)}} + \frac{7H\iota}{3N_h^k(s, a)} \tag{9}$$

holds for all $(s, a, h, k) \in S \times A \times [H] \times [K]$.

Event $\mathcal{E}^R$ means that the estimations of all reward functions stay in certain neighborhood of the true values. Event $\mathcal{E}^{PV}$ represents that the estimation of the value functions at the next step stay in some intervals. The following lemma shows $\mathcal{E}^R$ and $\mathcal{E}^{PV}$ hold with high probability. The analysis will be done assuming the successful event $\mathcal{E}^R \cap \mathcal{E}^{PV}$ holds in the rest of this section.

**Lemma 1** $\mathbb{P}(\mathcal{E}^R \cap \mathcal{E}^{PV}) \geq 1 - 3\delta$.

**Lemma 2** *Conditioned on* $\mathcal{E}^R \cap \mathcal{E}^{PV}$, $\overline{V}_h^k(s) \geq V_h^*(s) \geq V_h^{\overline{\pi}^k}(s) \geq \underline{V}_h^k(s)$ *and* $\overline{Q}_h^k(s, a) \geq Q_h^*(s, a) \geq Q_h^{\overline{\pi}^k}(s, a) \geq \underline{Q}_h^k(s, a)$ *for all* $(s, a, h, k) \in S \times A \times [H] \times [K]$.

## 6.2 REGRET ANALYSIS

We decompose the regret and analyze the different terms. Set $\Theta_h^k(s, a) = \sqrt{\frac{8 \mathbb{V}_{P_h} C_{h+1}^{\pi^{k*}, \underline{\pi}^k, \rho}(s, a) \iota}{N_h^k(s, a)}} + \sqrt{\frac{32}{N_h^k(s, a)}} + \frac{46\sqrt{SH^4\iota}}{N_h^k(s, a)}$, where $\pi^{k*}$ is the optimal policy towards the adversary policy $\underline{\pi}^k$ with $\pi_h^{k*}(s) = \arg\max_\pi C_h^{\pi, \underline{\pi}^k, \rho}(s)$. We define the cumulative regret of the output policy $\overline{\pi}^k$ at each episodes $k$ as $Regret(K) := \sum_{k=1}^K (V_1^*(s_1^k) - V_1^{\overline{\pi}^k}(s_1^k))$.

Let $M_1 = \sum_{k=1}^K \sum_{h=1}^H [\mathbb{D}_{\widetilde{\pi}_h^k} \hat{P}_h^k (\overline{V}_{h+1}^k - \underline{V}_{h+1}^k)(s_h^k) - \hat{P}_h^k(\overline{V}_{h+1}^k - \underline{V}_{h+1}^k)(s_h^k, a_h^k)]$, $M_2 = \sum_{k=1}^K \sum_{h=1}^H \frac{1}{H} [\mathbb{D}_{\widetilde{\pi}_h^k} P_h(\overline{V}_{h+1}^k - \underline{V}_{h+1}^k)(s_h^k) - P_h(\overline{V}_{h+1}^k - \underline{V}_{h+1}^k)(s_h^k, a_h^k)]$, $M_3 = \sum_{k=1}^K \sum_{h=1}^H (P_h^k(\overline{V}_{h+1}^k - \underline{V}_{h+1}^k)(s_h^k, a_h^k) - (\overline{V}_{h+1}^k - \underline{V}_{h+1}^k)(s_{h+1}^k))$ and $M_4 = \sum_{k=1}^K \sum_{h=1}^H [\frac{(SH + SH^2)\iota}{N_h^k(s_h^k, a_h^k)} + \mathbb{D}_{\widetilde{\pi}_h^k} \Theta_h^k(s_h^k)]$. Here $M_1$ and $M_2$ are the cumulative sample error from the random choices of the adversarial policy or agent's policy. $M_3$ is the cumulative sample error from the randomness of Monte Carlo sampling of the next state. $M_4$ is the cumulative error from the bonus item $\theta$. Lemma 3 shows that the regret can be bounded by these four terms.

**Lemma 3** *With probability at least* $1 - (S + 5)\delta$,

$$Regret(K) \leq \sum_{k=1}^K (\overline{V}_1^k(s_1^k) - \underline{V}_1^k(s_1^k)) \leq 21(M_1 + M_2 + M_3 + M_4). \tag{10}$$

We now bound each of these four items separately.

**Lemma 4** *With probability at least* $1 - \delta$, $|M_1| \leq H\sqrt{2HK\iota}$.

**Lemma 5** *With probability at least* $1 - \delta$, $|M_2| \leq \sqrt{2HK\iota}$.

**Lemma 6** *With probability at least* $1 - \delta$, $|M_3| \leq H\sqrt{2HK\iota}$.

**Lemma 7** *With probability at least* $1 - 2\delta$, $|M_4| \leq 2S^2AH^3\iota^2 + 8\sqrt{SAH^2K\iota} + 46S^{\frac{3}{2}}AH^3\iota^2 + \sqrt{24SAH^3K\iota} + 6\sqrt{SAH^5\iota}$.

**Putting all together.** By Lemmas 3, 4, 5, 6, and 7, we conclude that, with probability $1 - (S + 10)\delta$,

$$Regret(K) \leq O(\sqrt{H^3K\iota} + \sqrt{SAH^2K\iota} + \sqrt{SAH^3K\iota} + S^2AH^3\iota^2 + \sqrt{SAH^5\iota})$$
$$= O(\sqrt{SAH^3K\iota} + S^2AH^3\iota^2). \tag{11}$$

By rescaling $\delta$, $\log(\frac{2SAHK}{\delta/(S+10)}) \leq c\iota$ for some constant $c$ and we finish the proof of regret. As $\sum_{k=1}^{K}(\overline{V}_1^k(s_1^k) - \underline{V}_1^k(s_1^k)) \leq O(\sqrt{SAH^3K}\iota + S^2AH^3\iota^2)$, we have that $V_1^*(s_1) - V_1^{\pi^{out}}(s_1) \leq \min_k \overline{V}_1^k(s_1^k) - \underline{V}_1^k(s_1^k) \leq O(\frac{\sqrt{SAH^3\iota}}{K} + \frac{S^2AH^3\iota^2}{K})$ and we finish the proof of sample complexity.

## 7 SIMULATION RESULTS

We use OpenAI gym framework (Brockman et al., 2016), and consider two different problems: Cliff Walking, a toy text environment, and Inverted Pendulum, a control environment with the MuJoCo (Todorov et al., 2012) physics simulator. We set $H = 100$. To demonstrate the robustness, the policy is learned in a clean environment, and is then tested on the perturbed environment. Specifically, during the testing, we set a probability $p$ such that after the agent takes an action, with probability $p$, the action is chosen by an adversary. The adversary follows a fixed policy. A Monte-Carlo method is used to evaluate the accumulated reward of the learned policy on the perturbed environment. We take the average over 100 trajectories.

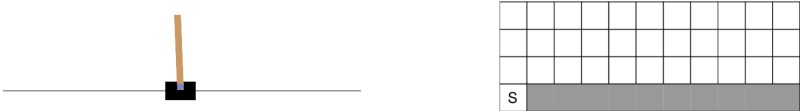

Figure 1: Inverted pendulum environment.          Figure 2: Cliff walking environment.

**Inverted pendulum.** The inverted pendulum experiment as shown in Figure 1 is a classic control problem in RL. An inverted pendulum is attached by a pivot point to a cart, which is restricted to linear movement in a plane. The cart can be pushed left or right, and the goal is to balance the inverted pendulum on the top of the cart by applying forces on the cart. A reward of $+1$ is awarded for each time step that the inverted pendulum stand upright within a certain angle limit. The fixed adversarial policy in the inverted pendulum environment is a force of $0.5$ N in the left direction.

**Cliff walking.** The cliff walking experiment as shown in Figure 2 is a classic scenario proposed in (Sutton & Barto, 2018). The game starts with the player at location $[3, 0]$ of the $4 \times 12$ grid world with the goal located at $[3, 11]$. A cliff runs along $[3, 1-10]$. If the player moves to a cliff location, it returns to the start location and receives a reward of $-100$. For every move which does not lead into the cliff, the agent receives a reward of $-1$. The player makes moves until they reach the goal. The fixed adversarial policy in the cliff walking environment is walking a step to the bottom.

To show the robustness, we compare our algorithm with a non-robust RL algorithm that is ORLC (Optimistic Reinforcement Learning with Certificates) in (Dann et al., 2019). We set $\rho = 0.2$ for our algorithm, which is the uncertain parameter used during the training. In Figure 3, we plot the accumulated reward of both algorithms under different $p$. It can be seen that overall our ARRLC algorithm achieves a much higher reward than the ORLC algorithm. This demonstrates the robustness of our ARRLC algorithm to policy execution uncertainty.

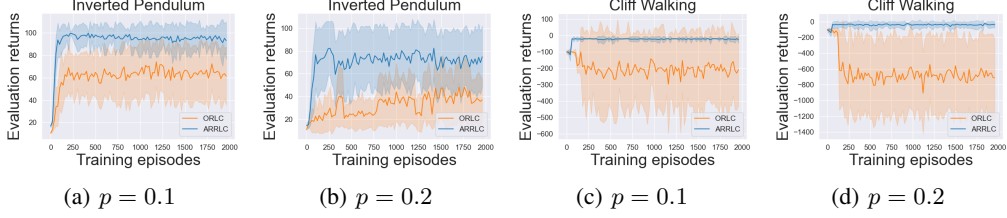

(a) $p = 0.1$          (b) $p = 0.2$          (c) $p = 0.1$          (d) $p = 0.2$

Figure 3: ARRLC v.s. ORLC (Dann et al., 2019)

To show the efficiency, we compare our algorithm with the robust TD algorithm in (Klima et al., 2019), which can converge to the optimal robust policy but has no theoretical guarantee on sample complexity or regret. We set $\rho = 0.2$. In Figure 4, we plot the accumulated reward of both algorithms

under different $p$ using a base-10 logarithmic scale on the x-axis and a linear scale on the y-axis. It can be seen that our ARRLC algorithm converges faster than the robust TD algorithm. This demonstrates the efficiency of our ARRLC algorithm to learn optimal policy under policy execution uncertainty.

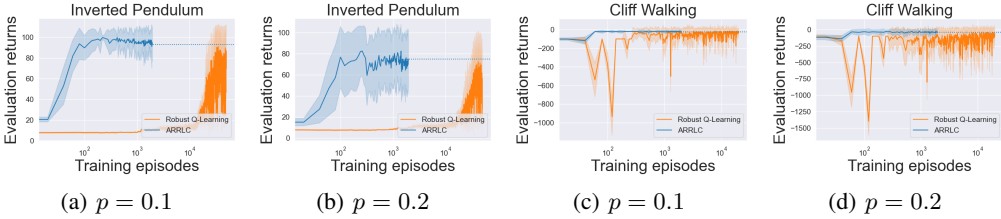

    (a) $p = 0.1$           (b) $p = 0.2$           (c) $p = 0.1$           (d) $p = 0.2$

Figure 4: ARRLC v.s. Robust TD (Klima et al., 2019)

We also compare our algorithm with the approaches in (Pinto et al., 2017; Tessler et al., 2019) that model the robust problem as a zero-sum game and alternating update the agent policy and adversary policy. In our implementation, (Pinto et al., 2017) fixes one policy and updates another for 25 episodes, then alternatively updates another in the next 25 episodes. (Tessler et al., 2019) does not alternate the updating until the current policy is converged. Figure 5 shows the efficiency of our ARRLC algorithm. ARRLC algorithm is more stable than the other algorithms.

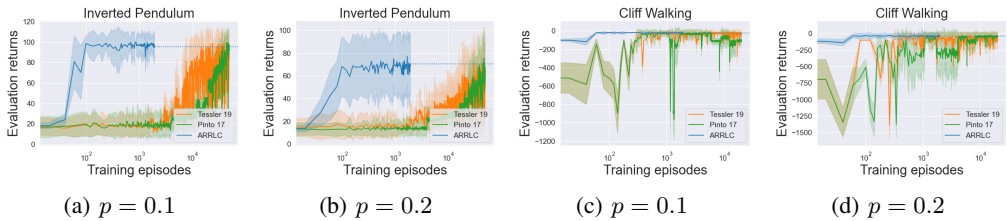

    (a) $p = 0.1$           (b) $p = 0.2$           (c) $p = 0.1$           (d) $p = 0.2$

Figure 5: ARRLC v.s. PR-PI (Tessler et al., 2019) v.s. RARL (Pinto et al., 2017)

We provide additional experimental results in Appendix A. We implement the ablation study by setting different $\rho$ and $p$ and try different adversary policies in the testing environment. We also perform cross-comparison experiments in which we use the learned worst-case policies to disturb the different robust agents.

## 8 CONCLUSION AND DISCUSSION

In this paper, we have developed a novel approach for solving action robust RL problems with probabilistic policy execution uncertainty. We have theoretically proved the sample complexity bound and the regret bound of the algorithms. The upper bound of the sample complexity and the regret of proposed ARRLC algorithm match the lower bound up to logarithmic factors, which shows the minimax optimality of our algorithm. Moreover, we have carried out numerical experiments to validate our algorithm's robustness and efficiency, revealing that ARRLC surpasses non-robust algorithms and converges more rapidly than the robust TD algorithm when faced with action perturbations.

The current theoretical guarantee on the sample complexity and regret of our algorithms are derived for the tabular setting. In the future work, we will explore action robust RL in continuous state or action space. Studying efficient action robust RL with function approximation is also an important direction to pursue. For this purpose, two insights from our work might be useful: (1) The adversary policy and the agent's policy can be simultaneously updated to efficiently sample trajectories; (2) The adversary policies at each episode do not necessarily need be the minimum over the actions, an approximation of the minimum also works. Based on these insights, a policy-gradient method could potentially be designed to handle the continuous action space. We could use policy gradient method, such like PPO, to find an approximation of the adversary policy (the minimum over actions). In addition, similar to (Zhou et al., 2023), considering a scalable uncertainty set is also an interesting direction.

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

# A    ADDITIONAL NUMERICAL RESULTS

## A.1    ABLATION STUDY AND MORE COMPARISONS

In Section 7, we compared our algorithm with the robust TD algorithm in (Klima et al., 2019). Here, we compare our algorithm with the algorithms in (Pinto et al., 2017; Tessler et al., 2019). The method in (Tessler et al., 2019) requires an MDP solver to solve the optimal adversarial policy when the agent policy is given and the optimal agent policy when the adversarial policy is given. The white-box MDP solver requires knowledge of the underline MDP so that there is no learning curve and sample complexity discussion in (Tessler et al., 2019). Thus, we implement the algorithms in (Pinto et al., 2017; Tessler et al., 2019) with a Q-learning MDP solver, and compared the final evaluation rewards and the learning curve. In addition, we implement the ablation study by setting different $\rho$ and $p$. In our experiments, the policy is learned in a clean environment, and is then tested on the perturbed environment. $\rho$ is the parameter in algorithm when learning the robust policy. $\rho$ can be considered as the agent's guess about the probability of a disturbance occurring. However, $p$ is the probability that the perturb happens in the perturbed environment. In the perturbed environment, with probability $p$, the action is perturbed by an adversarial action.

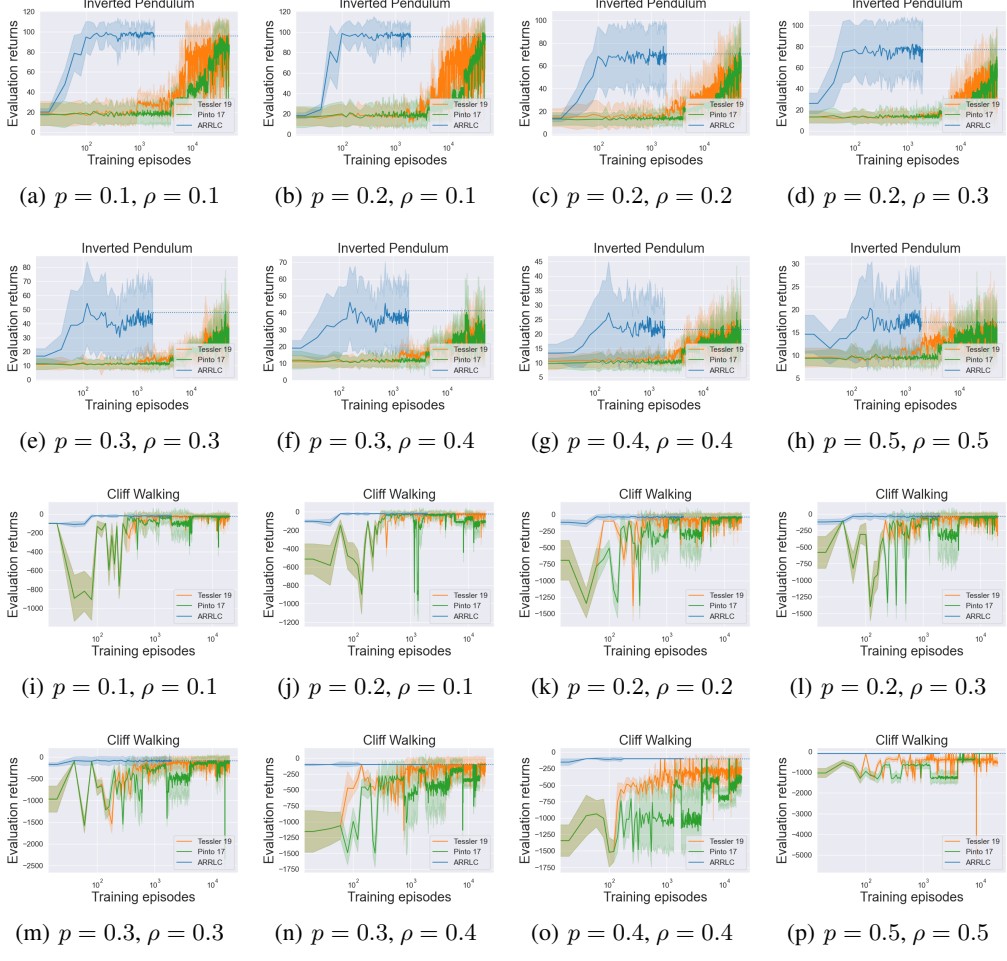

Figure 6: ARRLC v.s. RARL v.s. PR-PI

In Figure 6, we show the learning curves under different $p$ and $\rho$. It can be seen that our ARRLC algorithm converges faster than the other algorithms. This demonstrates the efficiency of our ARRLC algorithm to learn optimal policy under policy execution uncertainty.

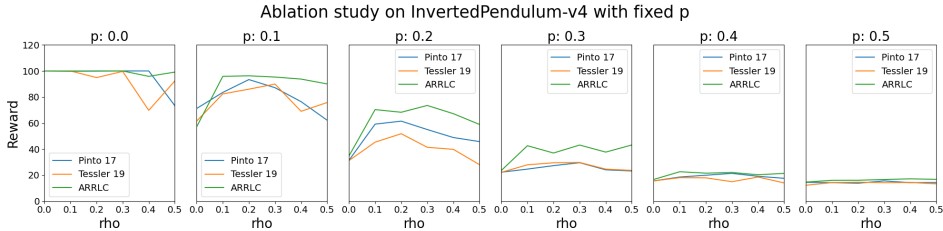

Figure 7: Ablation study on InvertedPendulum-v4 with fixed $\rho$.

In Figure 7, given the agents trained with fixed $rho$, we test the agents in different disturbed environments with different $p$. In Figure 8, we compared the different agents trained with different $rho$. The x-axis is the different choice of $\rho$ or $p$. The y-axis is the final evaluation rewards.

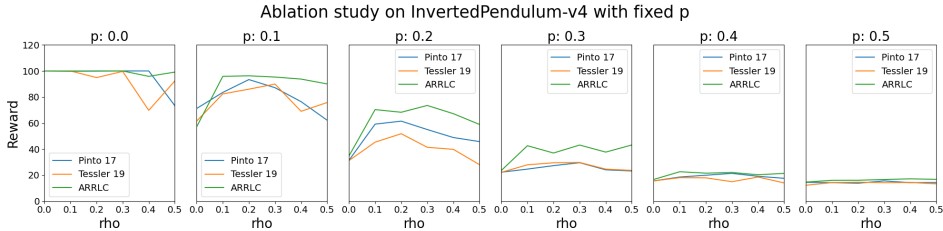

Figure 8: Ablation study on InvertedPendulum-v4 with fixed $\rho$.

The theoretical guarantee on sample complexity and regret of our algorithm relies on the assumption of known uncertainty parameter. However, in the experimental results shown in 7, the parameter can mismatch with the true disturb probability. In the main paper Figure 9, we test the mismatch of the uncertainty parameter $\rho$ and true uncertainty probability $p$. We trained the agent with $\rho = 0.2$, but we use $p = 0.1$ in the test. The proposed robust algorithm still outperforms the non-robust algorithm.

## A.2 ROBUSTNESS TO DIFFERENT ADVERSARY

In this section we considered different adversary policies include both the fixed policy in the main page and a random adversary policy. After the agent takes an action, with probability $p$, the random adversary will uniformly randomly choose an adversary action to replace the agent's action. In Figure 9 and Figure 10, "fix" represents that the actions are perturbed by a fixed adversarial policy during the testing, "random" represents that the actions are randomly perturbed during the testing, $p$ is the action perturbation probability.

Since we do not know whether the fixed policy or the random policy is the strongest adversary policy against the agent, a more direct comparison is to use the learned worst-case policy in different algorithms to do a cross-comparison. We used the learned worst-case policies to disturb the different robust agents. We report the final evaluation rewards in Table 1. We trained our method in 2000 episodes and the approaches of Pinto et al. (2017); Tessler et al. (2019) in 30000 episodes. We set that $p = \rho = 0.2$. The ARRLC agent performs the best against three different adversaries and the ARRLC adversary impacts the most on three different agents.

Table 1: Final rewards under cross-comparison between ARRLC, PR-PI and RAPL

|            | ARRLC adversary | RAPL adversary | PR-PI adversary |
| ---------- | --------------- | -------------- | --------------- |
| ARRLC agent | **72.536**     | **81.736**     | **89.824**      |
| RAPL agent  | 49.936         | 72.216         | 70.6            |
| PR-PI agent | 52.788         | 63.784         | 86.648          |

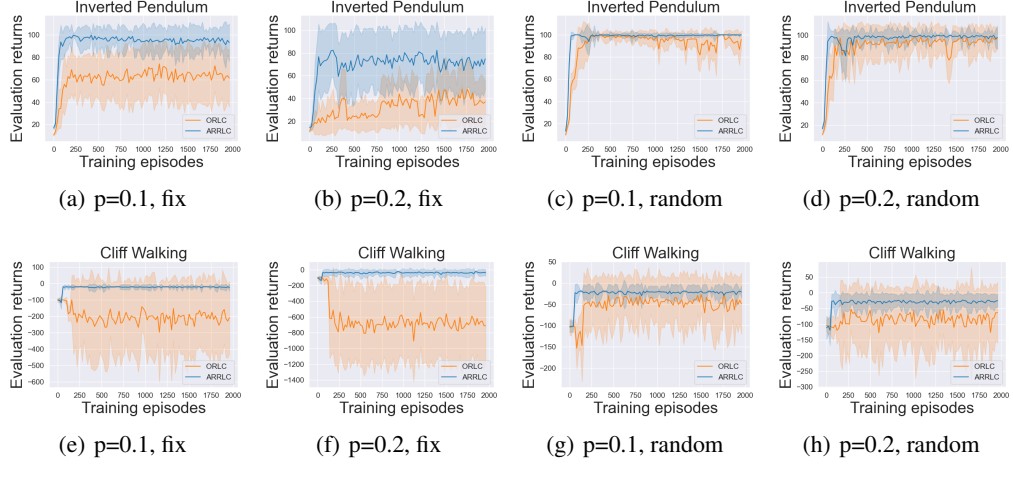

Figure 9: ARRLC v.s. ORLC.

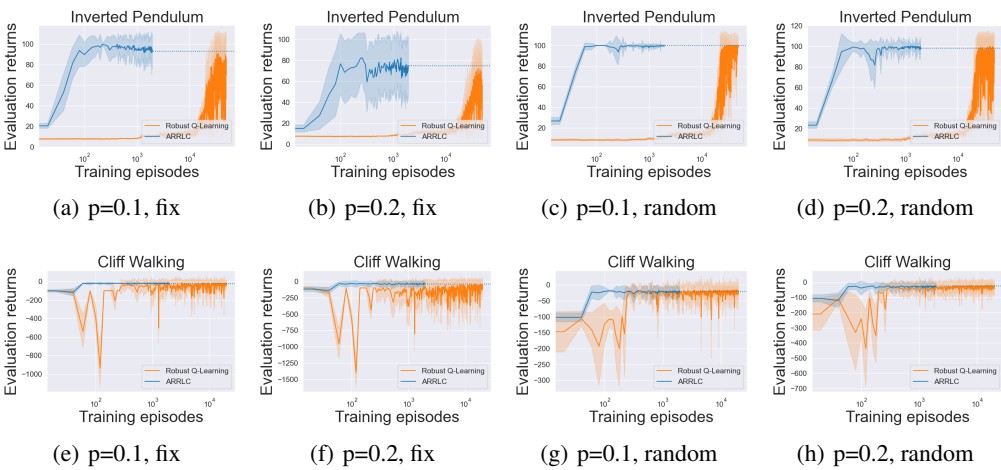

Figure 10: ARRLC v.s. Robust TD

## B PROOF OF PROPOSITION 1

The uncertainty set of the policy execution has the form in:

$$\Pi^\rho(\pi) := \{\widetilde{\pi} | \forall s, \widetilde{\pi}_h(\cdot|s) = (1-\rho)\pi(\cdot|s) + \rho\pi'_h(\cdot|s), \pi'_h(\cdot|s) \in \Delta_{\mathcal{A}}\}. \tag{12}$$

We define

$$
\begin{aligned}
C_h^{\pi,\pi',\rho}(s) &:= \mathbb{E}\left[\sum_{h'=h}^{H} R_{h'}(s_{h'}, a_{h'})|s_h = s, a_{h'} \sim \widetilde{\pi}_{h'}(\cdot|s_{h'})\right] \\
D_h^{\pi,\pi',\rho}(s,a) &:= \mathbb{E}\left[\sum_{h'=h}^{H} R_{h'}(s_{h'}, a_{h'})|s_h = s, a_h = a, a_{h'} \sim \widetilde{\pi}_{h'}(\cdot|s_{h'})\right].
\end{aligned}
$$

**Robust Bellman Equation**    First we prove the action robust Bellman equation holds for any policy $\pi$, state $s$ action $a$ and step $h$. From the definition of the robust value function in (1), we have $V_{H+1}^\pi(s) = 0, \forall s \in \mathcal{S}$.

We prove the robust Bellman equation by building a policy $\pi^-$. Here, policy $\pi^-$ is the optimal adversarial policy towards the policy $\pi$.

At step $H$, we set $\pi_H^-(s) = \arg\min_{a \in \mathcal{A}} R_H(s, a)$. We have

$$
\begin{aligned}
V_H^\pi(s) &= \min_{\pi'} C_H^{\pi, \pi', \rho}(s) \\
&= (1 - \rho)[\mathbb{D}_{\pi_H} R_H](s) + \rho \min_{\pi'}[\mathbb{D}_{\pi'_H} R_H](s) \\
&= (1 - \rho)[\mathbb{D}_{\pi_H} Q_H^\pi](s) + \rho \min_{a \in \mathcal{A}} Q_H^\pi(s, a) = C_H^{\pi, \pi^-, \rho}(s),
\end{aligned}
\tag{13}
$$

as $V_{H+1} = 0$.

The robust Bellman equation holds at step $H$ and $\min_{\pi'} \sum_s w(s) C_H^{\pi, \pi', \rho}(s) = \sum_s w(s) \min_{\pi'} C_H^{\pi, \pi', \rho}(s) = \sum_s w(s) C_H^{\pi, \pi^-, \rho}(s)$ for any state $s$ and any weighted function $w : \mathcal{S} \to \Delta_\mathcal{S}$.

Suppose the robust Bellman equation holds at step $h + 1$ and $\min_{\pi'} \sum_s w(s) C_{h+1}^{\pi, \pi', \rho}(s) = \sum_s w(s) \min_{\pi'} C_{h+1}^{\pi, \pi', \rho}(s) = \sum_s w(s) C_{h+1}^{\pi, \pi^-, \rho}(s)$ for any state $s$ and any weighted function $w : \mathcal{S} \to \Delta_\mathcal{S}$.

Now we prove the robust Bellman equation holds at step $h$. From the definition of the robust $Q$-function in (2) and the form of uncertainty set, we have

$$
\begin{aligned}
Q_h^\pi(s, a) &= \min_{\widetilde{\pi} \in \Pi(\pi)} \mathbb{E}\left[ \sum_{h'=h}^H R_{h'}(s_{h'}, a_{h'}) | s_h = s, a_h = a, a_{h'} \sim \widetilde{\pi}_{h'}(\cdot | s_{h'}) \right] \\
&= \min_{\pi'} D_h^{\pi, \pi', \rho}(s, a) \\
&= R_h(s, a) + \min_{\pi'} \mathbb{E}_{s' \sim P_h(\cdot | s, a)} C_{h+1}^{\pi, \pi', \rho}(s) \\
&= R_h(s, a) + \mathbb{E}_{s' \sim P_h(\cdot | s, a)} \min_{\pi'} C_{h+1}^{\pi, \pi', \rho}(s) \\
&= R_h(s, a) + [P_h V_{h+1}^\pi](s, a).
\end{aligned}
\tag{14}
$$

We also have that $Q_h^\pi(s, a) = D_h^{\pi, \pi^-, \rho}(s, a)$.

Recall that a (stochastic) Markov policy is a set of $H$ maps $\pi := \{\pi_h : \mathcal{S} \to \Delta_\mathcal{A}\}_{h \in [H]}$. From the definition of the robust value function in (1) and the form of uncertainty set, we have

$$
\begin{aligned}
V_h^\pi(s) &= \min_{\widetilde{\pi} \in \Pi(\pi)} \mathbb{E}\left[ \sum_{h'=h}^H R_{h'}(s_{h'}, a_{h'}) | s_h = s, a_{h'} \sim \widetilde{\pi}_{h'}(\cdot | s_{h'}) \right] \\
&= \min_{\pi'} C_h^{\pi, \pi', \rho}(s) \\
&= \min_{\pi'_h} \min_{\{\pi'_{h'}\}_{h'=h+1}^H} C_h^{\pi, \pi', \rho}(s) \\
&\geq (1 - \rho) \min_{\{\pi'_{h'}\}_{h'=h+1}^H} \mathbb{E}_{a \sim \pi_h(\cdot | s)} D_h^{\pi, \pi', \rho}(s, a) + \rho \min_{\pi'_h} \min_{\{\pi'_{h'}\}_{h'=h+1}^H} \mathbb{E}_{a \sim \pi'_h(\cdot | s)} D_h^{\pi, \pi', \rho}(s, a) \\
&\geq (1 - \rho) \mathbb{E}_{a \sim \pi_h(\cdot | s)} \min_{\{\pi'_{h'}\}_{h'=h+1}^H} D_h^{\pi, \pi', \rho}(s, a) + \rho \min_{\pi'_h} \mathbb{E}_{a \sim \pi'_h(\cdot | s)} \min_{\{\pi'_{h'}\}_{h'=h+1}^H} D_h^{\pi, \pi', \rho}(s, a) \\
&= (1 - \rho)[\mathbb{D}_{\pi_h} Q_h^\pi](s) + \rho \min_{a \in \mathcal{A}} Q_h^\pi(s, a).
\end{aligned}
\tag{15}
$$

We set $\pi_h^-(s) = \arg\min_{a \in \mathcal{A}} Q_h^\pi(s, a) = \arg\min_{a \in \mathcal{A}} D_h^{\pi, \pi^-, \rho}(s, a)$.

At step $h$, we have

$$
\begin{aligned}
V_h^\pi(s) &\leq C_h^{\pi, \pi^-, \rho}(s) \\
&= (1 - \rho)[\mathbb{D}_{\pi_h} D_h^{\pi, \pi^-, \rho}](s) + \rho \min_{a \in \mathcal{A}} D_h^{\pi, \pi^-, \rho}(s, a) \\
&= (1 - \rho)[\mathbb{D}_{\pi_h} Q_h^\pi](s) + \rho \min_{a \in \mathcal{A}} Q_h^\pi(s, a),
\end{aligned}
\tag{16}
$$

where the last equation comes from the robust Bellman equation at step $h + 1$ and

$$D_h^{\pi,\pi^-,\rho}(s,a) = R_h(s,a) + [P_h C_{h+1}^{\pi,\pi^-,\rho}](s,a) = R_h(s,a) + [P_h V_{h+1}^{\pi}](s,a).$$

Thus, the robust Bellman equation holds at step $h$.

Then, we prove the commutability of the expectation and the minimization operations at step $h$. For any weighted function $w$, we have $\min_{\pi'} \sum_s w(s) C_h^{\pi,\pi',\rho}(s) \geq \sum_s w(s) \min_{\pi'} C_h^{\pi,\pi',\rho}(s)$. Then, $\min_{\pi'} \sum_s w(s) C_h^{\pi,\pi',\rho}(s) \leq \sum_s w(s) C_h^{\pi,\pi^-,\rho}(s) = \sum_s w(s) \min_{\pi'} C_h^{\pi,\pi',\rho}(s)$.

By induction on $h = H, \cdots, 1$, we prove the robust Bellman equation.

**Perfect Duality and Robust Bellman Optimality Equation** We now prove that the perfect duality holds and can be solved by the optimal robust Bellman equation.

The control problem in the LHS of (4) is equivalent to

$$\max_{\pi} \min_{\widetilde{\pi} \in \Pi^\rho(\pi)} \mathbb{E}\left[\sum_{h'=h}^{H} R_{h'}(s_{h'}, a_{h'})|s_h = s, a_{h'} \sim \widetilde{\pi}_{h'}(\cdot|s_{h'})\right] = \max_{\pi} \min_{\pi'} C_h^{\pi,\pi',\rho}(s). \quad (17)$$

The control problem in the RHS of (4) is equivalent to

$$\min_{\widetilde{\pi} \in \Pi^\rho(\pi)} \max_{\pi} \mathbb{E}\left[\sum_{h'=h}^{H} R_{h'}(s_{h'}, a_{h'})|s_h = s, a_{h'} \sim \widetilde{\pi}_{h'}(\cdot|s_{h'})\right] = \min_{\pi'} \max_{\pi} C_h^{\pi,\pi',\rho}(s). \quad (18)$$

For step $H$, we have $C_H^{\pi,\pi',\rho}(s) = [\mathbb{D}_{((1-\rho)\pi+\rho\pi')_H} R_H](s) = (1-\rho)[\mathbb{D}_{\pi_H} R_H](s) + \rho[\mathbb{D}_{\pi'_H} R_H](s)$. Thus, we have

$$\begin{aligned}
\max_{\pi} \min_{\pi'} C_H^{\pi,\pi',\rho}(s) &= (1-\rho) \max_{\pi} [\mathbb{D}_{\pi_H} R_H](s) + \rho \min_{\pi'} [\mathbb{D}_{\pi'_H} R_H](s) \\
&= (1-\rho) \max_{a \in \mathcal{A}} R_H(s,a) + \rho \min_{b \in \mathcal{A}} R_H(s,b),
\end{aligned} \quad (19)$$

and

$$\begin{aligned}
\min_{\pi'} \max_{\pi} C_H^{\pi,\pi',\rho}(s) &= (1-\rho) \max_{\pi} [\mathbb{D}_{\pi_H} R_H](s) + \rho \min_{\pi'} [\mathbb{D}_{\pi'_H} R_H](s) \\
&= (1-\rho) \max_{a \in \mathcal{A}} R_H(s,a) + \rho \min_{b \in \mathcal{A}} R_H(s,b).
\end{aligned} \quad (20)$$

At step $H$, the perfect duality holds for all $s$ and there always exists an optimal robust policy $\pi_H^*(s) = \arg\max_{a \in \mathcal{A}} Q_H^*(s,a) = \arg\max_{a \in \mathcal{A}} R_H(s,a)$ and its corresponding optimal adversarial policy $\pi_H^-(s) = \arg\min_{a \in \mathcal{A}} R_H(s,a)$ which are deterministic. The action robust Bellman optimality equation holds at step $H$ for any stats $s$ and action $a$.

In addition, $\max_{\pi} \min_{\pi'} \sum_s w(s) C_H^{\pi,\pi',\rho}(s) = \sum_s w(s) \max_{\pi} \min_{\pi'} C_H^{\pi,\pi',\rho}(s)$ for any weighted function $w : \mathcal{S} \to \Delta_{\mathcal{S}}$. This can be shown as

$$\begin{aligned}
&\max_{\pi} \min_{\pi'} \sum_{s \in \mathcal{S}} w(s) C_H^{\pi,\pi',\rho}(s) \\
&= (1-\rho) \max_{\pi} \sum_{s \in \mathcal{S}} w(s) [\mathbb{D}_{\pi_H} R_H](s) + \rho \min_{\pi'} \sum_{s \in \mathcal{S}} w(s) [\mathbb{D}_{\pi'_H} R_H](s) \\
&= (1-\rho) \sum_{s \in \mathcal{S}} w(s) \max_{a \in \mathcal{A}} R_H(s,a) + \rho \sum_{s \in \mathcal{S}} w(s) \min_{b \in \mathcal{A}} R_H(s,b).
\end{aligned} \quad (21)$$

Suppose that at steps from $h + 1$ to $H$, the perfect duality holds for any $s$, the action robust Bellman optimality equation holds for any state $s$ and action $a$, there always exists an optimal robust policy $\pi_{h'}^* = \arg\max_{a \in \mathcal{A}} Q_{h'}^*(s,a)$ and its corresponding optimal adversarial policy $\pi_{h'}^-(s) = \arg\min_{a \in \mathcal{A}} Q_{h'}^*(s,a), \forall h' \geq h + 1$, which is deterministic, and $\max_{\pi} \min_{\pi'} \sum_s w(s) C_{h'}^{\pi,\pi^-,\rho}(s) = \sum_s w(s) \max_{\pi} \min_{\pi'} C_{h'}^{\pi,\pi',\rho}(s)$ for any state $s$, any weighted function $w : \mathcal{S} \to \Delta_{\mathcal{S}}$ and any

$h' \geq h+1$. We have $V_{h'}^*(s) = V_{h'}^{\pi^*}(s) = C_{h'}^{\pi^*,\pi^-,\rho}(s)$ and $Q_{h'}^*(s,a) = Q_{h'}^{\pi^*}(s,a) = D_{h'}^{\pi^*,\pi^-,\rho}(s,a)$ for any state $s$ and any $h' \geq h+1$.

We first prove that the robust Bellman optimality equation holds at step $h$.

We have

$$
\begin{aligned}
Q_h^*(s,a) &= \max_\pi \min_{\pi'} D_h^{\pi,\pi',\rho}(s,a) \\
&= \max_\pi \min_{\pi'} (R_h(s,a) + [P_h C_{h+1}^{\pi,\pi',\rho}](s,a)) \\
&= R_h(s,a) + [P_h(\max_\pi \min_{\pi'} C_{h+1}^{\pi,\pi',\rho})](s,a) \\
&= R_h(s,a) + [P_h V_{h+1}^*](s,a).
\end{aligned}
\tag{22}
$$

and also $Q_h^*(s,a) = Q_h^{\pi^*}(s,a) = D_h^{\pi^*,\pi^-,\rho}(s,a)$.

From the robust Bellman equation, we have

$$
\begin{aligned}
\max_\pi V_h^\pi(s) =& \max_\pi \left( (1-\rho)[\mathbb{D}_{\pi_h} Q_h^\pi](s) + \rho \min_{a \in \mathcal{A}} Q_h^\pi(s,a) \right) \\
\leq& (1-\rho) \max_{\pi_h} \max_{\{\pi_h\}_{h'=h+1}^H} [\mathbb{D}_{\pi_h} Q_h^\pi](s) + \rho \max_{\{\pi_h\}_{h'=h+1}^H} \min_{a \in \mathcal{A}} Q_h^\pi(s,a) \\
\leq& (1-\rho) \max_{\pi_h} \max_{\{\pi_h\}_{h'=h+1}^H} [\mathbb{D}_{\pi_h} Q_h^\pi](s) + \rho \min_{a \in \mathcal{A}} \max_{\{\pi_h\}_{h'=h+1}^H} Q_h^\pi(s,a) \\
\leq& (1-\rho) \max_{\pi_h} [\mathbb{D}_{\pi_h} Q_h^*](s) + \rho \min_{a \in \mathcal{A}} Q_h^*(s,a) \\
=& (1-\rho) \max_{a \in \mathcal{A}} Q_h^*(s,a) + \rho \min_{a \in \mathcal{A}} Q_h^*(s,a).
\end{aligned}
\tag{23}
$$

We set $\pi_h^*(s) = \max_{a \in \mathcal{A}} Q_h^*(s,a)$. According to the robust bellman equation, we have

$$
\begin{aligned}
\max_\pi V_h^\pi(s) \geq V_h^{\pi^*}(s) &= (1-\rho)[\mathbb{D}_{\pi_h^*} Q_h^{\pi^*}](s) + \rho \min_{a \in \mathcal{A}} Q_h^{\pi^*}(s,a) \\
&= (1-\rho) \max_{a \in \mathcal{A}} Q_h^{\pi^*}(s,a) + \rho \min_{a \in \mathcal{A}} Q_h^{\pi^*}(s,a) \\
&= (1-\rho) \max_{a \in \mathcal{A}} Q_h^*(s,a) + \rho \min_{a \in \mathcal{A}} Q_h^*(s,a).
\end{aligned}
\tag{24}
$$

Thus, the robust Bellman optimality equation holds at step $h$. There always exists an optimal robust policy $\pi_h^* = \arg\max_{a \in \mathcal{A}} Q_h^*(s,a)$ and its corresponding optimal adversarial policy $\pi_h^-(s) = \arg\min_{a \in \mathcal{A}} Q_h^*(s,a)$ that is deterministic so that $C_h^{\pi^*,\pi^-,\rho}(s) = V_h^*(s)$.

Then, we prove the commutability of the expectation, the minimization and the maximization operations at step $h$.

In the proof of robust Bellman equation, we have shown that

$$
\min_{\pi'} \sum_s w(s) C_h^{\pi,\pi',\rho}(s) = \sum_s w(s) \min_{\pi'} C_h^{\pi,\pi',\rho}(s)
$$

for any policy $\pi$ and any weighted function $w$. Hence

$$
\max_\pi \min_{\pi'} \sum_s w(s) C_h^{\pi,\pi',\rho}(s) \sum_s = \max_\pi \sum_s w(s) \min_{\pi'} C_h^{\pi,\pi',\rho}(s).
$$

First, we have

$$
\max_\pi \sum_s w(s) \min_{\pi'} C_h^{\pi,\pi',\rho}(s) \leq \sum_s w(s) \max_\pi \min_{\pi'} C_h^{\pi,\pi',\rho}(s).
$$

Then, we can show

$$
\begin{aligned}
\max_\pi \sum_s w(s) \min_{\pi'} C_h^{\pi,\pi',\rho}(s) &\geq \sum_s w(s) \min_{\pi'} C_h^{\pi^*,\pi',\rho}(s) \\
&= \sum_s w(s) C_h^{\pi^*,\pi^-,\rho}(s) \\
&= \sum_s w(s) \max_\pi \min_{\pi'} C_h^{\pi,\pi',\rho}(s).
\end{aligned}
\tag{25}
$$

In summary,

$$\max_{\pi} \min_{\pi'} \sum_s w(s) C_h^{\pi,\pi',\rho}(s) \sum_s = w(s) \max_{\pi} \min_{\pi'} C_h^{\pi,\pi',\rho}(s).$$

We can show the perfect duality at step $h$ by

$$\max_{\pi} \min_{\pi'} C_h^{\pi,\pi',\rho}(s) = C_h^{\pi^*,\pi^-,\rho}(s) = \max_{\pi} C_h^{\pi,\pi^-,\rho}(s) \geq \min_{\pi'} \max_{\pi} C_h^{\pi,\pi',\rho}(s). \qquad (26)$$

By induction on $h = H, \cdots, 1$, we prove Proposition 1.

## C  PROOF FOR ACTION ROBUST REINFORCEMENT LEARNING WITH CERTIFICATES

In this section, we prove Theorem 1. Recall that we use $\overline{Q}_h^k, \overline{V}_h^k, \underline{Q}_h^k, \underline{V}_h^k$, $N_h^k$, $\hat{P}_h^k, \hat{r}_h^k$ and $\theta_h^k$ to denote the values of $\overline{Q}_h, \overline{V}_h, \underline{Q}_h, \underline{V}_h$, $\max\{N_h, 1\}$, $\hat{P}_h$, $r_h$ and $\theta_h$ at the beginning of the $k$-th episode in Algorithm 1.

### C.1  PROOF OF MONOTONICITY

#### C.1.1  PROOF OF LEMMA 1

When $N_h^k(s,a) \leq 1$, (8), (9) and (7) hold trivially by the bound of the rewards and value functions.

For every $h \in [H]$ the empiric Bernstein inequality combined with a union bound argument, to take into account that $N_h^k(s,a) > 1$ is a random number, leads to the following inequality w.p. $1 - SAH\delta$ (see Theorem 4 in (Maurer & Pontil, 2009))

$$\left| (\hat{P}_h^k - P_h) V_{h+1}^*(s,a) \right| \leq \sqrt{\frac{2\mathbb{V}_{\hat{P}_h^k} V_{h+1}^*(s,a)\iota}{N_h^k(s,a)}} + \frac{7H\iota}{3(N_h^k(s,a))}, \qquad (27)$$

and

$$\left| (\hat{P}_h^k - P_h) V_{h+1}^{\overline{\pi}^k}(s,a) \right| \leq \sqrt{\frac{2\mathbb{V}_{\hat{P}_h^k} V_{h+1}^{\overline{\pi}^k}(s,a)\iota}{N_h^k(s,a)}} + \frac{7H\iota}{3(N_h^k(s,a))}. \qquad (28)$$

Similarly, with Azuma's inequality, w.p. $1 - SAH\delta$

$$\left| \hat{r}_h^k(s,a) - R_h(s,a) \right| \leq \sqrt{\frac{2Var(r_h^k(s,a))\iota}{N_h^k(s,a)}} + \frac{7\iota}{3(N_h^k(s,a))} \leq \sqrt{\frac{2\hat{r}_h^k(s,a)\iota}{N_h^k(s,a)}} + \frac{7\iota}{3(N_h^k(s,a))}, \qquad (29)$$

where $Var(r_h^k(s,a))$ is the empirical variance of $R_h(s,a)$ computed by the $N_h^k(s,a)$ samples and $Var(r_h^k(s,a)) \leq \hat{r}_h^k(s,a)$ .

#### C.1.2  PROOF OF LEMMA 2

We first prove that $\overline{Q}_h^k(s,a) \geq Q_h^*(s,a)$ for all $(s,a,h,k) \in S \times A \times [H] \times [K]$, by backward induction conditioned on the event $\mathcal{E}^R \cap \mathcal{E}^{PV}$. Firstly, the conclusion holds for $h = H + 1$ because $\overline{V}_{H+1}(s) = \underline{V}_{H+1}(s) = 0$ and $\overline{Q}_{H+1}(s,a) = \underline{Q}_{H+1}(s,a) = 0$ for all $s$ and $a$. For $h \in [H]$,

assuming the conclusion holds for $h+1$, by Algorithm 1, we have

$$\hat{r}_h^k(s,a) + \hat{P}_h^k \overline{V}_{h+1}(s,a) + \theta_h^k(s,a) - Q_h^*(s,a)$$

$$=\hat{r}_h^k(s,a) + \hat{P}_h^k \overline{V}_{h+1}(s,a) + \theta_h^k(s,a) - R_h(s,a) - P_h V_{h+1}^*(s,a)$$

$$=\hat{r}_h^k(s,a) - R_h(s,a) + \hat{P}_h^k \left( \overline{V}_{h+1} - V_{h+1}^* \right)(s,a) + (\hat{P}_h^k - P_h)V_{h+1}^*(s,a) + \theta_h^k(s,a)$$

$$\geq (\hat{P}_h^k - P_h)V_{h+1}^*(s,a) + \sqrt{\frac{2\mathbb{V}_{\hat{P}_h^k}[(\overline{V}_{h+1}^k + \underline{V}_{h+1}^k)/2](s,a)\iota}{N_h^k(s,a)}} + \frac{\hat{P}_h^k \left( \overline{V}_{h+1}^k - \underline{V}_{h+1}^k \right)(s,a)}{H} + \frac{8H^2\iota}{N_h^k(s,a)}$$

$$\geq \sqrt{\frac{2\mathbb{V}_{\hat{P}_h^k}[(\overline{V}_{h+1}^k + \underline{V}_{h+1}^k)/2](s,a)\iota}{N_h^k(s,a)}} + \frac{\hat{P}_h^k \left( \overline{V}_{h+1}^k - \underline{V}_{h+1}^k \right)(s,a)}{H} + \frac{8H^2\iota}{N_h^k(s,a)} - \sqrt{\frac{2\mathbb{V}_{\hat{P}_h^k}V_{h+1}^*(s,a)\iota}{N_h^k(s,a)}},$$
$$\tag{30}$$

where the first inequality comes from event $\mathcal{E}^R$, $\overline{V}_{h+1}(s) \geq V_{h+1}^*(s)$ and the definition of $\theta_h^k(s,a)$ and the last inequality from event $\mathcal{E}^{PV}$. By the relation of $V$-values in the step $(h+1)$,

$$\left| \mathbb{V}_{\hat{P}_h^k} \left( \frac{\overline{V}_{h+1}^k + \underline{V}_{h+1}^k}{2} \right)(s,a) - \mathbb{V}_{\hat{P}_h^k} V_{h+1}^*(s,a) \right|$$

$$\leq \left| [\hat{P}_h^k(\overline{V}_{h+1}^k + \underline{V}_{h+1}^k)/2]^2 - (\hat{P}_h^k V_{h+1}^*)^2 \right|(s,a) + \left| \hat{P}_h^k[(\overline{V}_{h+1}^k + \underline{V}_{h+1}^k)/2]^2 - \hat{P}_h^k(V_{h+1}^*)^2 \right|(s,a)$$

$$\leq 4H\hat{P}_h^k \left| (\overline{V}_{h+1}^k + \underline{V}_{h+1}^k)/2 - V_{h+1}^* \right|(s,a)$$

$$\leq 2H\hat{P}_h^k \left( \overline{V}_{h+1}^k - \underline{V}_{h+1}^k \right)(s,a)$$
$$\tag{31}$$

and

$$\sqrt{\frac{2\mathbb{V}_{\hat{P}_h^k}V_{h+1}^*(s,a)\iota}{N_h^k(s,a)}}$$

$$\leq \sqrt{\frac{2\mathbb{V}_{\hat{P}_h^k}[(\overline{V}_{h+1}^k + \underline{V}_{h+1}^k)/2](s,a)\iota + 4H\hat{P}_h^k \left( \overline{V}_{h+1}^k - \underline{V}_{h+1}^k \right)(s,a)\iota}{N_h^k(s,a)}}$$

$$\leq \sqrt{\frac{2\mathbb{V}_{\hat{P}_h^k}[(\overline{V}_{h+1}^k + \underline{V}_{h+1}^k)/2](s,a)\iota}{N_h^k(s,a)}} + \sqrt{\frac{4H\hat{P}_h^k \left( \overline{V}_{h+1}^k - \underline{V}_{h+1}^k \right)(s,a)\iota}{N_h^k(s,a)}}$$

$$\leq \sqrt{\frac{2\mathbb{V}_{\hat{P}_h^k}[(\overline{V}_{h+1}^k + \underline{V}_{h+1}^k)/2](s,a)\iota}{N_h^k(s,a)}} + \frac{\hat{P}_h^k \left( \overline{V}_{h+1}^k - \underline{V}_{h+1}^k \right)(s,a)}{H} + \frac{8H^2\iota}{N_h^k(s,a)}.$$
$$\tag{32}$$

Plugging (32) back into (30), we have $\hat{r}_h^k(s,a) + \hat{P}_h^k \overline{V}_{h+1}(s,a) + \theta_h^k(s,a) \geq Q_h^*(s,a)$. Thus, $\overline{Q}_h^k(s,a) = \min\{H - h + 1, \hat{r}_h^k(s,a) + \hat{P}_h^k \overline{V}_{h+1}^k(s,a) + \theta_h^k(s,a)\} \geq Q_h^*(s,a)$.

From the definition of $\overline{V}_h^k(s)$ and $\overline{\pi}_h^k$, we have

$$\begin{aligned}
\overline{V}_h^k(s) &= (1-\rho)\overline{Q}_h^k(s, \overline{\pi}_h^k(s)) + \rho \overline{Q}_h^k(s, \underline{\pi}_h^k(s)) \\
&\geq (1-\rho)\overline{Q}_h^k(s, \pi_h^*(s)) + \rho Q_h^*(s, \underline{\pi}_h^k(s)) \\
&\geq (1-\rho)Q_h^*(s, \pi_h^*(s)) + \rho \min_{a \in \mathcal{A}} Q_h^*(s,a) = V_h^*(s).
\end{aligned}$$
$$\tag{33}$$

Similarly, we can prove that $\underline{Q}_h^k(s,a) \leq Q_h^{\overline{\pi}^k}(s,a)$ and $\underline{V}_h^k(s) \leq V_h^{\overline{\pi}^k}(s)$.

$$\hat{r}_h^k(s,a) + \hat{P}_h^k \underline{V}_{h+1}(s,a) - \theta_h^k(s,a) - Q_h^{\overline{\pi}^k}(s,a)$$

$$=\hat{r}_h^k(s,a) + \hat{P}_h^k \underline{V}_{h+1}(s,a) - \theta_h^k(s,a) - R_h(s,a) - P_h V_{h+1}^{\overline{\pi}^k}(s,a)$$

$$=\hat{r}_h^k(s,a) - R_h(s,a) + \hat{P}_h^k \left( \underline{V}_{h+1} - V_{h+1}^{\overline{\pi}^k} \right)(s,a) + (\hat{P}_h^k - P_h)V_{h+1}^{\overline{\pi}^k}(s,a) - \theta_h^k(s,a)$$

$$\leq (\hat{P}_h^k - P_h)V_{h+1}^{\overline{\pi}^k}(s,a) - \sqrt{\frac{2\mathbb{V}_{\hat{P}_h^k}[(\overline{V}_{h+1}^k + \underline{V}_{h+1}^k)/2](s,a)\iota}{N_h^k(s,a)}}$$

$$- \frac{\hat{P}_h^k \left( \overline{V}_{h+1}^k - \underline{V}_{h+1}^k \right)(s,a)}{H} - \frac{8H^2\iota}{N_h^k(s,a)} \tag{34}$$

$$\leq \sqrt{\frac{2\mathbb{V}_{\hat{P}_h^k}V_{h+1}^{\overline{\pi}^k}(s,a)\iota}{N_h^k(s,a)}} - \sqrt{\frac{2\mathbb{V}_{\hat{P}_h^k}[(\overline{V}_{h+1}^k + \underline{V}_{h+1}^k)/2](s,a)\iota}{N_h^k(s,a)}}$$

$$- \frac{\hat{P}_h^k \left( \overline{V}_{h+1}^k - \underline{V}_{h+1}^k \right)(s,a)}{H} - \frac{8H^2\iota}{N_h^k(s,a)} \leq 0,$$

and

$$\underline{V}_h^k(s) = (1-\rho)\underline{Q}_h^k(s, \overline{\pi}_h^k(s)) + \rho \underline{Q}_h^k(s, \underline{\pi}_h^k(s))$$

$$\leq (1-\rho)Q_h^{\overline{\pi}^k}(s, \overline{\pi}_h^k(s)) + \rho \min_{a \in \mathcal{A}} \underline{Q}_h^k(s,a)$$

$$\leq (1-\rho)Q_h^{\overline{\pi}^k}(s, \overline{\pi}_h^k(s)) + \rho \underline{Q}_h^k(s, \arg\min_{a \in \mathcal{A}} Q_h^{\overline{\pi}^k}(s,a)) \tag{35}$$

$$\leq (1-\rho)Q_h^{\overline{\pi}^k}(s, \overline{\pi}_h^k(s)) + \rho \min_{a \in \mathcal{A}} Q_h^{\overline{\pi}^k}(s,a) = V_h^{\overline{\pi}^k}(s).$$

## C.2 REGRET ANALYSIS

### C.2.1 PROOF OF LEMMA 3

We consider the event $\mathcal{E}^R \cap \mathcal{E}^{PV}$. The following analysis will be done assuming the successful event $\mathcal{E}^R \cap \mathcal{E}^{PV}$ holds. By Lemma 2, the regret can be bounded by $Regret(K) := \sum_{k=1}^K (V_1^*(s_1^k) - V_1^{\overline{\pi}^k}(s_1^k)) \leq \sum_{k=1}^K (\overline{V}_1^k(s_1^k) - \underline{V}_1^k(s_1^k))$.

By the update steps in Algorithm 1, we have

$$
\overline{V}_h^k(s_h^k) - \underline{V}_h^k(s_h^k)
$$

$$
=(1-\rho)\overline{Q}_h^k(s_h^k, \overline{\pi}_h^k(s_h^k)) + \rho\overline{Q}_h^k(s_h^k, \underline{\pi}_h^k(s_h^k)) - (1-\rho)\underline{Q}_h^k(s_h^k, \overline{\pi}_h^k(s_h^k)) - \rho\underline{Q}_h^k(s_h^k, \underline{\pi}_h^k(s_h^k))
$$

$$
\leq [\mathbb{D}_{\widetilde{\pi}_h^k}\hat{P}_h^k(\overline{V}_{h+1}^k - \underline{V}_{h+1}^k)](s_h^k) + 2\mathbb{D}_{\widetilde{\pi}_h^k}\theta_h(s_h^k)
$$

$$
=[\mathbb{D}_{\widetilde{\pi}_h^k}\hat{P}_h^k(\overline{V}_{h+1}^k - \underline{V}_{h+1}^k)](s_h^k) - [\hat{P}_h^k(\overline{V}_{h+1}^k - \underline{V}_{h+1}^k)](s_h^k, a_h^k) + 2\mathbb{D}_{\widetilde{\pi}_h^k}\theta_h(s_h^k)
$$

$$
\quad + [\hat{P}_h^k(\overline{V}_{h+1}^k - \underline{V}_{h+1}^k)](s_h^k, a_h^k)
$$

$$
=[\mathbb{D}_{\widetilde{\pi}_h^k}\hat{P}_h^k(\overline{V}_{h+1}^k - \underline{V}_{h+1}^k)](s_h^k) - [\hat{P}_h^k(\overline{V}_{h+1}^k - \underline{V}_{h+1}^k)](s_h^k, a_h^k) + 2\mathbb{D}_{\widetilde{\pi}_h^k}\theta_h(s_h^k)
$$

$$
\quad + [\hat{P}_h^k(\overline{V}_{h+1}^k - \underline{V}_{h+1}^k)](s_h^k, a_h^k) - c_1 P_h(\overline{V}_{h+1}^k - \underline{V}_{h+1}^k)(s_h^k, a_h^k)
$$

$$
\quad + c_1 P_h(\overline{V}_{h+1}^k - \underline{V}_{h+1}^k)(s_h^k, a_h^k) - c_2(\overline{V}_{h+1}^k - \underline{V}_{h+1}^k)(s_{h+1}^k) + c_2(\overline{V}_{h+1}^k - \underline{V}_{h+1}^k)(s_{h+1}^k)
$$

$$
=[\mathbb{D}_{\widetilde{\pi}_h^k}\hat{P}_h^k(\overline{V}_{h+1}^k - \underline{V}_{h+1}^k)](s_h^k) - [\hat{P}_h^k(\overline{V}_{h+1}^k - \underline{V}_{h+1}^k)](s_h^k, a_h^k)
$$

$$
\quad + [\hat{P}_h^k(\overline{V}_{h+1}^k - \underline{V}_{h+1}^k)](s_h^k, a_h^k) - c_1 P_h(\overline{V}_{h+1}^k - \underline{V}_{h+1}^k)(s_h^k, a_h^k)
$$

$$
\quad + c_1 P_h(\overline{V}_{h+1}^k - \underline{V}_{h+1}^k)(s_h^k, a_h^k) - c_2(\overline{V}_{h+1}^k - \underline{V}_{h+1}^k)(s_{h+1}^k) + c_2(\overline{V}_{h+1}^k - \underline{V}_{h+1}^k)(s_{h+1}^k)
$$

$$
\quad + 2(1-\rho)\sqrt{\frac{2\mathbb{V}_{\hat{P}_h^k}[(\overline{V}_{h+1}^k + \underline{V}_{h+1}^k)/2](s_h^k, \overline{\pi}_h^k(s_h^k))\iota}{N_h^k(s_h^k, \overline{\pi}_h^k(s_h^k))}} + 2(1-\rho)\sqrt{\frac{2\hat{r}_h^k(s_h^k, \overline{\pi}_h^k(s_h^k))\iota}{N_h^k(s_h^k, \overline{\pi}_h^k(s_h^k))}}
$$

$$
\quad + (1-\rho)\hat{P}_h^k(\overline{V}_{h+1}^k - \underline{V}_{h+1}^k)(s_h^k, \overline{\pi}_h^k(s_h^k))/H + \frac{2(1-\rho)(24H^2 + 7H + 7)\iota}{3N_h^k(s_h^k, \overline{\pi}_h^k(s_h^k))}
$$

$$
\quad + 2\rho\sqrt{\frac{2\mathbb{V}_{\hat{P}_h^k}[(\overline{V}_{h+1}^k + \underline{V}_{h+1}^k)/2](s_h^k, \underline{\pi}_h^k(s_h^k))\iota}{N_h^k(s_h^k, \underline{\pi}_h^k(s_h^k))}} + 2\rho\sqrt{\frac{2\hat{r}_h^k(s_h^k, \underline{\pi}_h^k(s_h^k))\iota}{N_h^k(s_h^k, \underline{\pi}_h^k(s_h^k))}}
$$

$$
\quad + \rho\hat{P}_h^k(\overline{V}_{h+1}^k - \underline{V}_{h+1}^k)(s_h^k, \underline{\pi}_h^k(s_h^k))/H + \frac{2\rho(24H^2 + 7H + 7)\iota}{3N_h^k(s_h^k, \underline{\pi}_h^k(s_h^k))}
$$

$$
=(1+1/H)[\mathbb{D}_{\widetilde{\pi}_h^k}\hat{P}_h^k(\overline{V}_{h+1}^k - \underline{V}_{h+1}^k)](s_h^k) - (1+1/H)[\hat{P}_h^k(\overline{V}_{h+1}^k - \underline{V}_{h+1}^k)](s_h^k, a_h^k)
$$

$$
\quad + \underbrace{(1+1/H)[\hat{P}_h^k(\overline{V}_{h+1}^k - \underline{V}_{h+1}^k)](s_h^k, a_h^k) - c_1 P_h(\overline{V}_{h+1}^k - \underline{V}_{h+1}^k)(s_h^k, a_h^k)}_{(a)}
$$

$$
\quad + c_1 P_h(\overline{V}_{h+1}^k - \underline{V}_{h+1}^k)(s_h^k, a_h^k) - c_2(\overline{V}_{h+1}^k - \underline{V}_{h+1}^k)(s_{h+1}^k) + c_2(\overline{V}_{h+1}^k - \underline{V}_{h+1}^k)(s_{h+1}^k)
$$

$$
\quad + 2(1-\rho)\underbrace{\sqrt{\frac{2\mathbb{V}_{\hat{P}_h^k}[(\overline{V}_{h+1}^k + \underline{V}_{h+1}^k)/2](s_h^k, \overline{\pi}_h^k(s_h^k))\iota}{N_h^k(s_h^k, \overline{\pi}_h^k(s_h^k))}}}_{(b1)} + 2(1-\rho)\sqrt{\frac{2\hat{r}_h^k(s_h^k, \overline{\pi}_h^k(s_h^k))\iota}{N_h^k(s_h^k, \overline{\pi}_h^k(s_h^k))}}
$$

$$
\quad + \frac{2(1-\rho)(24H^2 + 7H + 7)\iota}{3N_h^k(s_h^k, \overline{\pi}_h^k(s_h^k))} + 2\rho\underbrace{\sqrt{\frac{2\mathbb{V}_{\hat{P}_h^k}[(\overline{V}_{h+1}^k + \underline{V}_{h+1}^k)/2](s_h^k, \underline{\pi}_h^k(s_h^k))\iota}{N_h^k(s_h^k, \underline{\pi}_h^k(s_h^k))}}}_{(b2)}
$$

$$
\quad + 2\rho\sqrt{\frac{2\hat{r}_h^k(s_h^k, \underline{\pi}_h^k(s_h^k))\iota}{N_h^k(s_h^k, \underline{\pi}_h^k(s_h^k))}} + \frac{2\rho(24H^2 + 7H + 7)\iota}{3N_h^k(s_h^k, \underline{\pi}_h^k(s_h^k))}.
$$

(36)

**Bound of the error of the empirical probability estimator (a)** By Bennett's inequality, we have that w.p. $1 - S\delta$

$$
|\hat{P}_h^k(s'|s,a) - P_h(s'|s,a)| \leq \sqrt{\frac{2P_h(s'|s,a)\iota}{N_h^k(s,a)}} + \frac{\iota}{3N_h^k(s,a)} \tag{37}
$$

holds for all $s, a, h, k, s'$.

Thus, we have that

$$
\begin{aligned}
&(\hat{P}_h^k - P_h)(\overline{V}_{h+1}^k - \underline{V}_{h+1}^k)(s,a) \\
&= \sum_{s'} (\hat{P}_h^k(s'|s,a) - P_h(s'|s,a))(\overline{V}_{h+1}^k(s') - \underline{V}_{h+1}^k(s')) \\
&\leq \sum_{s'} \sqrt{\frac{2P_h(s'|s,a)\iota}{N_h^k(s,a)}}(\overline{V}_{h+1}^k(s') - \underline{V}_{h+1}^k(s')) + \frac{SH\iota}{3N_h^k(s,a)} \\
&\leq \sum_{s'} \left( \frac{P_h(s'|s,a)\iota}{H} + \frac{H}{2N_h^k(s,a)} \right) \left( \overline{V}_{h+1}^k(s') - \underline{V}_{h+1}^k(s') \right) + \frac{SH\iota}{3N_h^k(s,a)} \\
&\leq P_h(\overline{V}_{h+1}^k - \underline{V}_{h+1}^k)(s,a)/H + \frac{SH^2}{2N_h^k(s,a)} + \frac{SH\iota}{3N_h^k(s,a)} \\
&\leq P_h(\overline{V}_{h+1}^k - \underline{V}_{h+1}^k)(s,a)/H + \frac{SH^2\iota}{N_h^k(s,a)},
\end{aligned}
\tag{38}
$$

where the second inequality is due to AM-GM inequality.

**Bound of the error of the empirical variance estimator (b1) & (b2)** Here, we bound $\mathbb{V}_{\hat{P}_h^k}[(\overline{V}_{h+1}^k + \underline{V}_{h+1}^k)/2](s_h^k, a_h^k)$.

Recall that $C_h^{\pi, \pi', \rho}(s) = \mathbb{E}\left[ \sum_{h'=h}^H R_{h'}(s_{h'}, a_{h'}) | s_h = s, a_{h'} \sim \widetilde{\pi}_{h'}(\cdot|s_{h'}) \right]$ in Appendix B. Set $\pi^{k*}$ here is the optimal policy towards the adversary policy $\underline{\pi}^k$ with $\pi_h^{k*}(s) = \arg\max_\pi C_h^{\pi, \underline{\pi}^k, \rho}(s)$. Similar to the proof in Appendix C.1.2, we can show that $\overline{V}_h^k(s) \geq C_h^{\pi^{k*}, \underline{\pi}^k, \rho}(s)$. We also have that $C_h^{\pi^{k*}, \underline{\pi}^k, \rho}(s) = \max_\pi C_h^{\pi, \underline{\pi}^k, \rho}(s) \geq C_h^{\overline{\pi}^k, \underline{\pi}^k, \rho}(s) \geq V_h^{\overline{\pi}^k}(s) \geq \underline{V}_h^k(s)$. For any $(s, a, h, k) \in \mathcal{S} \times \mathcal{A} \times [H] \times [K]$, under event $\mathcal{E}^R \cap \mathcal{E}^{PV}$,

$$
\begin{aligned}
&\mathbb{V}_{\hat{P}_h^k}[(\overline{V}_{h+1}^k + \underline{V}_{h+1}^k)/2](s,a) - \mathbb{V}_{P_h} C_{h+1}^{\pi^{k*}, \underline{\pi}^k, \rho}(s,a) \\
&= \hat{P}_h^k[(\overline{V}_{h+1}^k + \underline{V}_{h+1}^k)/2]^2(s,a) - [\hat{P}_h^k(\overline{V}_{h+1}^k + \underline{V}_{h+1}^k)/2]^2(s,a) \\
&\quad - P_h(C_{h+1}^{\pi^{k*}, \underline{\pi}^k, \rho})^2(s,a) + (P_h C_{h+1}^{\pi^{k*}, \underline{\pi}^k, \rho})^2(s,a) \\
&\leq [\hat{P}_h^k(\overline{V}_{h+1}^k)^2 - (\hat{P}_h^k \underline{V}_{h+1}^k)^2 - P_h(\underline{V}_{h+1}^k)^2 + (P_h \overline{V}_{h+1}^k)^2](s,a) \\
&\leq |(\hat{P}_h^k - P_h)(\overline{V}_{h+1}^k)^2|(s,a) + |(P_h \underline{V}_{h+1}^k)^2 - (\hat{P}_h^k \underline{V}_{h+1}^k)^2|(s,a) \\
&\quad + P_h|(\overline{V}_{h+1}^k)^2 - (\underline{V}_{h+1}^k)^2|(s,a) + |(P_h \overline{V}_{h+1}^k)^2 - (P_h \underline{V}_{h+1}^k)^2|(s,a),
\end{aligned}
\tag{39}
$$

where the first inequality is due $\overline{V}_h^k(s) \geq C_h^{\pi^{k*}, \underline{\pi}^k, \rho}(s) \geq \underline{V}_h^k(s)$. The result of (Weissman et al., 2003) combined with a union bound on $N_h^k(s,a) \in [K]$ implies w.p $1 - \delta$

$$
\|\hat{P}_h^k(\cdot|s,a) - P_h(\cdot|s,a)\|_1 \leq \sqrt{\frac{2S\iota}{N_h^k(s,a)}}
\tag{40}
$$

holds for all $s, a, h, k$.

These terms can be bounded separately by

$$
|(\hat{P}_h^k - P_h)(\overline{V}_{h+1}^k)^2|(s,a) \leq H^2 \sqrt{\frac{2S\iota}{N_h^k(s,a)}},
$$

$$
|(P_h \underline{V}_{h+1}^k)^2 - (\hat{P}_h^k \underline{V}_{h+1}^k)^2|(s,a) \leq 2H|(P_h - \hat{P}_h^k)\underline{V}_{h+1}^k| \leq 2H^2 \sqrt{\frac{2S\iota}{N_h^k(s,a)}},
\tag{41}
$$

$$
P_h|(\overline{V}_{h+1}^k)^2 - (\underline{V}_{h+1}^k)^2|(s,a) \leq 2H P_h(\overline{V}_{h+1}^k - \underline{V}_{h+1}^k)(s,a),
$$

$$
|(P_h \overline{V}_{h+1}^k)^2 - (P_h \underline{V}_{h+1}^k)^2|(s,a) \leq 2H P_h(\overline{V}_{h+1}^k - \underline{V}_{h+1}^k)(s,a),
$$

where the first two inequality is due to (40). In addition, $3H^2\sqrt{\frac{2S\iota}{N_h^k(s,a)}} \leq 1 + \frac{9SH^4\iota}{2N_h^k(s,a)}$. Thus, we have

$$
\begin{aligned}
&(1-\rho)\sqrt{\frac{\mathbb{V}_{\hat{P}_h^k}[(\overline{V}_{h+1}^k + \underline{V}_{h+1}^k)/2](s_h^k, \overline{\pi}_h^k(s_h^k))\iota}{N_h^k(s_h^k, \overline{\pi}_h^k(s_h^k))}} + \rho\sqrt{\frac{\mathbb{V}_{\hat{P}_h^k}[(\overline{V}_{h+1}^k + \underline{V}_{h+1}^k)/2](s_h^k, \underline{\pi}_h^k(s_h^k))\iota}{N_h^k(s_h^k, \underline{\pi}_h^k(s_h^k))}} \\
&\leq (1-\rho)\sqrt{\frac{\mathbb{V}_{P_h} C_{h+1}^{\pi^{k*}, \underline{\pi}^k, \rho}(s_h^k, \overline{\pi}_h^k(s_h^k))\iota}{N_h^k(s_h^k, \overline{\pi}_h^k(s_h^k))}} + \rho\sqrt{\frac{\mathbb{V}_{P_h} C_{h+1}^{\pi^{k*}, \underline{\pi}^k, \rho}(s_h^k, \underline{\pi}_h^k(s_h^k))\iota}{N_h^k(s_h^k, \underline{\pi}_h^k(s_h^k))}} \\
&\quad + (1-\rho)\sqrt{\frac{4HP_h(\overline{V}_{h+1}^k - \underline{V}_{h+1}^k)(s_h^k, \overline{\pi}_h^k(s_h^k))\iota}{N_h^k(s_h^k, \overline{\pi}_h^k(s_h^k))}} + \rho\sqrt{\frac{4HP_h(\overline{V}_{h+1}^k - \underline{V}_{h+1}^k)(s_h^k, \underline{\pi}_h^k(s_h^k))\iota}{N_h^k(s_h^k, \underline{\pi}_h^k(s_h^k))}} \\
&\quad + (1-\rho)\sqrt{\frac{1}{N_h^k(s_h^k, \overline{\pi}_h^k(s_h^k))}} + \rho\sqrt{\frac{1}{N_h^k(s_h^k, \underline{\pi}_h^k(s_h^k))}} + \frac{(1-\rho)\sqrt{9SH^4\iota/2}}{N_h^k(s_h^k, \overline{\pi}_h^k(s_h^k))} + \frac{\rho\sqrt{9SH^4\iota/2}}{N_h^k(s_h^k, \underline{\pi}_h^k(s_h^k))} \\
&\leq (1-\rho)\sqrt{\frac{\mathbb{V}_{P_h} C_{h+1}^{\pi^{k*}, \underline{\pi}^k, \rho}(s_h^k, \overline{\pi}_h^k(s_h^k))\iota}{N_h^k(s_h^k, \overline{\pi}_h^k(s_h^k))}} + \rho\sqrt{\frac{\mathbb{V}_{P_h} C_{h+1}^{\pi^{k*}, \underline{\pi}^k, \rho}(s_h^k, \underline{\pi}_h^k(s_h^k))\iota}{N_h^k(s_h^k, \underline{\pi}_h^k(s_h^k))}} \\
&\quad + (1-\rho)\left(\frac{P_h(\overline{V}_{h+1}^k - \underline{V}_{h+1}^k)(s_h^k, \overline{\pi}_h^k(s_h^k))}{2\sqrt{2}H} + \frac{2\sqrt{2}H^2\iota}{N_h^k(s_h^k, \overline{\pi}_h^k(s_h^k))}\right) \\
&\quad + \rho\left(\frac{P_h(\overline{V}_{h+1}^k - \underline{V}_{h+1}^k)(s_h^k, \underline{\pi}_h^k(s_h^k))}{2\sqrt{2}H} + \frac{2\sqrt{2}H^2\iota}{N_h^k(s_h^k, \underline{\pi}_h^k(s_h^k))}\right) \\
&\quad + (1-\rho)\sqrt{\frac{1}{N_h^k(s_h^k, \overline{\pi}_h^k(s_h^k))}} + \rho\sqrt{\frac{1}{N_h^k(s_h^k, \underline{\pi}_h^k(s_h^k))}} \\
&\quad + \frac{(1-\rho)\sqrt{9SH^4\iota/2}}{N_h^k(s_h^k, \overline{\pi}_h^k(s_h^k))} + \frac{\rho\sqrt{9SH^4\iota/2}}{N_h^k(s_h^k, \underline{\pi}_h^k(s_h^k))} \\
&= (1-\rho)\sqrt{\frac{\mathbb{V}_{P_h} C_{h+1}^{\pi^{k*}, \underline{\pi}^k, \rho}(s_h^k, \overline{\pi}_h^k(s_h^k))\iota}{N_h^k(s_h^k, \overline{\pi}_h^k(s_h^k))}} + \rho\sqrt{\frac{\mathbb{V}_{P_h} C_{h+1}^{\pi^{k*}, \underline{\pi}^k, \rho}(s_h^k, \underline{\pi}_h^k(s_h^k))\iota}{N_h^k(s_h^k, \underline{\pi}_h^k(s_h^k))}} \\
&\quad + \frac{\mathbb{D}_{\widetilde{\pi}_h^k} P_h(\overline{V}_{h+1}^k - \underline{V}_{h+1}^k)(s_h^k)}{2\sqrt{2}H} + \frac{2\sqrt{2}(1-\rho)H^2\iota}{N_h^k(s_h^k, \overline{\pi}_h^k(s_h^k))} + \frac{2\sqrt{2}\rho H^2\iota}{N_h^k(s_h^k, \underline{\pi}_h^k(s_h^k))} \\
&\quad + (1-\rho)\sqrt{\frac{1}{N_h^k(s_h^k, \overline{\pi}_h^k(s_h^k))}} + \rho\sqrt{\frac{1}{N_h^k(s_h^k, \underline{\pi}_h^k(s_h^k))}} \\
&\quad + \frac{(1-\rho)\sqrt{9SH^4\iota/2}}{N_h^k(s_h^k, \overline{\pi}_h^k(s_h^k))} + \frac{\rho\sqrt{9SH^4\iota/2}}{N_h^k(s_h^k, \underline{\pi}_h^k(s_h^k))},
\end{aligned}
\tag{42}
$$

where the second inequality is due to AM-GM inequality.

**Recursing on** $h$ Plugging (38) and (42) into (36)and setting $c_1 = 1 + 1/H$ and $c_2 = (1 + 1/H)^3$ , we have

$$
\overline{V}_h^k(s_h^k) - \underline{V}_h^k(s_h^k)
$$

$$
\leq (1 + 1/H)[\mathbb{D}_{\widetilde{\pi}_h^k}\hat{P}_h^k(\overline{V}_{h+1}^k - \underline{V}_{h+1}^k)](s_h^k) - (1 + 1/H)[\hat{P}_h^k(\overline{V}_{h+1}^k - \underline{V}_{h+1}^k)](s_h^k, a_h^k)
$$

$$
+ (1/H + 1/H^2)P_h(\overline{V}_{h+1}^k - \underline{V}_{h+1}^k)(s_h^k, a_h^k) + \frac{(SH + SH^2)\iota}{N_h^k(s_h^k, a_h^k)}
$$

$$
+ c_1 P_h(\overline{V}_{h+1}^k - \underline{V}_{h+1}^k)(s_h^k, a_h^k) - c_2(\overline{V}_{h+1}^k - \underline{V}_{h+1}^k)(s_{h+1}^k) + c_2(\overline{V}_{h+1}^k - \underline{V}_{h+1}^k)(s_{h+1}^k)
$$

$$
+ 2(1 - \rho)\sqrt{\frac{2\hat{r}_h^k(s_h^k, \overline{\pi}_h^k(s_h^k))\iota}{N_h^k(s_h^k, \overline{\pi}_h^k(s_h^k))}} + \frac{2(1 - \rho)(24H^2 + 7H + 7)\iota}{3N_h^k(s_h^k, \overline{\pi}_h^k(s_h^k)))}
$$

$$
+ 2\rho\sqrt{\frac{2\hat{r}_h^k(s_h^k, \underline{\pi}_h^k(s_h^k))\iota}{N_h^k(s_h^k, \underline{\pi}_h^k(s_h^k))}} + \frac{2\rho(24H^2 + 7H + 7)\iota}{3N_h^k(s_h^k, \underline{\pi}_h^k(s_h^k)))}
$$

$$
+ (1 - \rho)\sqrt{\frac{8\mathbb{V}_{P_h}C_{h+1}^{\pi^{k*}, \underline{\pi}^k, \rho}(s_h^k, \overline{\pi}_h^k(s_h^k))\iota}{N_h^k(s_h^k, \overline{\pi}_h^k(s_h^k))}} + \rho\sqrt{\frac{8\mathbb{V}_{P_h}C_{h+1}^{\pi^{k*}, \underline{\pi}^k, \rho}(s_h^k, \underline{\pi}_h^k(s_h^k))\iota}{N_h^k(s_h^k, \underline{\pi}_h^k(s_h^k))}}
$$

$$
+ \frac{\mathbb{D}_{\widetilde{\pi}_h^k}P_h(\overline{V}_{h+1}^k - \underline{V}_{h+1}^k)(s_h^k)}{H} + \frac{8(1 - \rho)H^2\iota}{N_h^k(s_h^k, \overline{\pi}_h^k(s_h^k))} + \frac{8\rho H^2\iota}{N_h^k(s_h^k, \underline{\pi}_h^k(s_h^k))}
$$

$$
+ (1 - \rho)\sqrt{\frac{8}{N_h^k(s_h^k, \overline{\pi}_h^k(s_h^k))}} + \rho\sqrt{\frac{8}{N_h^k(s_h^k, \underline{\pi}_h^k(s_h^k))}} + \frac{6(1 - \rho)\sqrt{SH^4\iota}}{N_h^k(s_h^k, \overline{\pi}_h^k(s_h^k))} + \frac{6\rho\sqrt{SH^4\iota}}{N_h^k(s_h^k, \underline{\pi}_h^k(s_h^k))}.
$$

$$
(43)
$$

We set $\Theta_h^k(s, a) = \sqrt{\frac{8\mathbb{V}_{P_h}C_{h+1}^{\pi^{k*}, \underline{\pi}^k, \rho}(s,a)\iota}{N_h^k(s,a)}} + \sqrt{\frac{32}{N_h^k(s,a)}} + \frac{46\sqrt{SH^4\iota}}{N_h^k(s,a)}$. Since $r_h^k(s, a) \leq 1$, by organizing the items, we have that

$$
\overline{V}_h^k(s_h^k) - \underline{V}_h^k(s_h^k)
$$

$$
\leq (1 + 1/H)[\mathbb{D}_{\widetilde{\pi}_h^k}\hat{P}_h^k(\overline{V}_{h+1}^k - \underline{V}_{h+1}^k)](s_h^k) - (1 + 1/H)[\hat{P}_h^k(\overline{V}_{h+1}^k - \underline{V}_{h+1}^k)](s_h^k, a_h^k)
$$

$$
+ (1/H + 1/H^2)P_h(\overline{V}_{h+1}^k - \underline{V}_{h+1}^k)(s_h^k, a_h^k) + \frac{(SH + SH^2)\iota}{N_h^k(s_h^k, a_h^k)}
$$

$$
+ c_1 P_h(\overline{V}_{h+1}^k - \underline{V}_{h+1}^k)(s_h^k, a_h^k) - c_2(\overline{V}_{h+1}^k - \underline{V}_{h+1}^k)(s_{h+1}^k) + c_2(\overline{V}_{h+1}^k - \underline{V}_{h+1}^k)(s_{h+1}^k)
$$

$$
+ \frac{\mathbb{D}_{\widetilde{\pi}_h^k}P_h(\overline{V}_{h+1}^k - \underline{V}_{h+1}^k)(s_h^k, \overline{\pi}_h^k(s_h^k))}{H} + \mathbb{D}_{\widetilde{\pi}_h^k}\Theta_h^k(s_h^k)
$$

$$
\leq (1 + 1/H)[\mathbb{D}_{\widetilde{\pi}_h^k}\hat{P}_h^k(\overline{V}_{h+1}^k - \underline{V}_{h+1}^k)](s_h^k) - (1 + 1/H)[\hat{P}_h^k(\overline{V}_{h+1}^k - \underline{V}_{h+1}^k)](s_h^k, a_h^k)
$$

$$
+ \frac{1}{H}[\mathbb{D}_{\widetilde{\pi}_h^k}P_h(\overline{V}_{h+1}^k - \underline{V}_{h+1}^k)(s_h^k) - P_h(\overline{V}_{h+1}^k - \underline{V}_{h+1}^k)(s_h^k, a_h^k)]
$$

$$
+ (1 + 3/H + 1/H^2)P_h(\overline{V}_{h+1}^k - \underline{V}_{h+1}^k)(s_h^k, a_h^k) - c_2(\overline{V}_{h+1}^k - \underline{V}_{h+1}^k)(s_{h+1}^k)
$$

$$
+ c_2(\overline{V}_{h+1}^k - \underline{V}_{h+1}^k)(s_{h+1}^k) + \frac{(SH + SH^2)\iota}{N_h^k(s_h^k, a_h^k)} + \mathbb{D}_{\widetilde{\pi}_h^k}\Theta_h^k(s_h^k)
$$

$$
\leq (1 + 1/H)[\mathbb{D}_{\widetilde{\pi}_h^k}\hat{P}_h^k(\overline{V}_{h+1}^k - \underline{V}_{h+1}^k)](s_h^k) - (1 + 1/H)[\hat{P}_h^k(\overline{V}_{h+1}^k - \underline{V}_{h+1}^k)](s_h^k, a_h^k)
$$

$$
+ \frac{1}{H}[\mathbb{D}_{\widetilde{\pi}_h^k}P_h(\overline{V}_{h+1}^k - \underline{V}_{h+1}^k)(s_h^k) - P_h(\overline{V}_{h+1}^k - \underline{V}_{h+1}^k)(s_h^k, a_h^k)]
$$

$$
+ c_2 P_h(\overline{V}_{h+1}^k - \underline{V}_{h+1}^k)(s_h^k, a_h^k) - c_2(\overline{V}_{h+1}^k - \underline{V}_{h+1}^k)(s_{h+1}^k)
$$

$$
+ c_2(\overline{V}_{h+1}^k - \underline{V}_{h+1}^k)(s_{h+1}^k) + \frac{(SH + SH^2)\iota}{N_h^k(s_h^k, a_h^k)} + \mathbb{D}_{\widetilde{\pi}_h^k}\Theta_h^k(s_h^k).
$$

By induction of (36) on $h = 1, \cdots, H$ and $\overline{V}_{h+1}^k = \underline{V}_{h+1}^k = 0$, we have that

$$
\begin{aligned}
Regret(K) \leq 21 \sum_{k=1}^{K} \sum_{h=1}^{H} &(\mathbb{D}_{\widetilde{\pi}_h^k} \hat{P}_h^k (\overline{V}_{h+1}^k - \underline{V}_{h+1}^k)(s_h^k) - \hat{P}_h^k(\overline{V}_{h+1}^k - \underline{V}_{h+1}^k)(s_h^k, a_h^k) \\
&+ \frac{1}{H}[\mathbb{D}_{\widetilde{\pi}_h^k} P_h (\overline{V}_{h+1}^k - \underline{V}_{h+1}^k)(s_h^k) - P_h(\overline{V}_{h+1}^k - \underline{V}_{h+1}^k)(s_h^k, a_h^k)] \\
&+ P_h(\overline{V}_{h+1}^k - \underline{V}_{h+1}^k)(s_h^k, a_h^k) - (\overline{V}_{h+1}^k - \underline{V}_{h+1}^k)(s_{h+1}^k) \\
&+ \frac{(SH + SH^2)\iota}{N_h^k(s_h^k, a_h^k)} + \mathbb{D}_{\widetilde{\pi}_h^k} \Theta_h^k(s_h^k)).
\end{aligned}
\tag{45}
$$

Here we use $(1 + 1/H)^{3H} < 21$.

### C.2.2 PROOF OF LEMMA 4

Recall that $M_1 = \sum_{k=1}^{K} \sum_{h=1}^{H} [\mathbb{D}_{\widetilde{\pi}_h^k} \hat{P}_h^k (\overline{V}_{h+1}^k - \underline{V}_{h+1}^k)(s_h^k) - \hat{P}_h^k(\overline{V}_{h+1}^k - \underline{V}_{h+1}^k)(s_h^k, a_h^k)]$.

Since $\mathbb{E}_{a_h^k \sim \mathbb{D}_{\widetilde{\pi}_h^k}} [\hat{P}_h^k(\overline{V}_{h+1}^k - \underline{V}_{h+1}^k)(s_h^k, a_h^k)] = \mathbb{D}_{\widetilde{\pi}_h^k} \hat{P}_h^k (\overline{V}_{h+1}^k - \underline{V}_{h+1}^k)(s_h^k)$, we have that $\mathbb{D}_{\widetilde{\pi}_h^k} \hat{P}_h^k (\overline{V}_{h+1}^k - \underline{V}_{h+1}^k)(s_h^k) - \hat{P}_h^k(\overline{V}_{h+1}^k - \underline{V}_{h+1}^k)(s_h^k, a_h^k)$ is a martingale difference sequence. By the Azuma-Hoeffding inequality, with probability $1 - \delta$, we have

$$
\left| \sum_{k=1}^{K} \sum_{h=1}^{H} [\mathbb{D}_{\widetilde{\pi}_h^k} \hat{P}_h^k (\overline{V}_{h+1}^k - \underline{V}_{h+1}^k)(s_h^k) - \hat{P}_h^k(\overline{V}_{h+1}^k - \underline{V}_{h+1}^k)(s_h^k, a_h^k)] \right| \leq H\sqrt{2HK\iota}.
\tag{46}
$$

### C.2.3 PROOF OF LEMMA 5

Recall that $M_2 = \sum_{k=1}^{K} \sum_{h=1}^{H} \frac{1}{H}[\mathbb{D}_{\widetilde{\pi}_h^k} P_h (\overline{V}_{h+1}^k - \underline{V}_{h+1}^k)(s_h^k) - P_h(\overline{V}_{h+1}^k - \underline{V}_{h+1}^k)(s_h^k, a_h^k)]$.

Since $\mathbb{E}_{a_h^k \sim \mathbb{D}_{\widetilde{\pi}_h^k}} [P_h(\overline{V}_{h+1}^k - \underline{V}_{h+1}^k)(s_h^k, a_h^k)] = \mathbb{D}_{\widetilde{\pi}_h^k} P_h (\overline{V}_{h+1}^k - \underline{V}_{h+1}^k)(s_h^k)$, we have that $\mathbb{D}_{\widetilde{\pi}_h^k} P_h (\overline{V}_{h+1}^k - \underline{V}_{h+1}^k)(s_h^k) - P_h(\overline{V}_{h+1}^k - \underline{V}_{h+1}^k)(s_h^k, a_h^k)$ is a martingale difference sequence. By the Azuma-Hoeffding inequality, with probability $1 - \delta$, we have

$$
\left| \sum_{k=1}^{K} \sum_{h=1}^{H} [\mathbb{D}_{\widetilde{\pi}_h^k} P_h (\overline{V}_{h+1}^k - \underline{V}_{h+1}^k)(s_h^k) - P_h(\overline{V}_{h+1}^k - \underline{V}_{h+1}^k)(s_h^k, a_h^k)] \right| \leq H\sqrt{2HK\iota}.
\tag{47}
$$

### C.2.4 PROOF OF LEMMA 6

Recall that $M_3 = \sum_{k=1}^{K} \sum_{h=1}^{H} (P_h^k(\overline{V}_{h+1}^k - \underline{V}_{h+1}^k)(s_h^k, a_h^k) - (\overline{V}_{h+1}^k - \underline{V}_{h+1}^k)(s_{h+1}^k))$.

Let the one-hot vector $\hat{\mathbb{1}}_h^k(\cdot | s_h^k, a_h^k)$ to satisfy that $\hat{\mathbb{1}}_h^k(s_{h+1}^k | s_h^k, a_h^k) = 1$ and $\hat{\mathbb{1}}_h^k(s | s_h^k, a_h^k) = 0$ for $s \neq s_{h+1}^k$. Thus, $[(P_h^k - \hat{\mathbb{1}}_h^k)(\overline{V}_{h+1}^k - \underline{V}_{h+1}^k)](s_h^k, a_h^k)$ is a martingale difference sequence. By the Azuma-Hoeffding inequality, with probability $1 - \delta$, we have

$$
\left| \sum_{k=1}^{K} \sum_{h=1}^{H} [(P_h^k - \hat{\mathbb{1}}_h^k)(\overline{V}_{h+1}^k - \underline{V}_{h+1}^k)](s_h^k, a_h^k) \right| \leq H\sqrt{2HK\iota}.
\tag{48}
$$

### C.2.5 PROOF OF LEMMA 7

We bounded $M_4 = \sum_{k=1}^{K} \sum_{h=1}^{H} [\frac{(SH+SH^2)\iota}{N_h^k(s_h^k, a_h^k)} + \mathbb{D}_{\widetilde{\pi}_h^k} \Theta_h^k(s_h^k)]$ by separately bounding the four items.

**Bound** $\sum_{k=1}^{K} \sum_{h=1}^{H} \frac{(SH+SH^2)\iota}{N_h^k(s_h^k, a_h^k)}$   We regroup the summands in a different way.

$$\sum_{k=1}^{K} \sum_{h=1}^{H} \frac{(SH+SH^2)\iota}{N_h^k(s_h^k, a_h^k)} = (SH+SH^2)\iota \sum_{h=1}^{H} \sum_{(s,a)\in\mathcal{S}\times\mathcal{A}} \sum_{n=1}^{N_h^K(s,a)} \frac{1}{n} \leq (SH+SH^2)SAH\iota^2.$$

(49)

Recall that $\Theta_h^k(s,a) = \sqrt{\frac{8\mathbb{V}_{P_h} C_{h+1}^{\pi^{k*}, \underline{\pi}^k, \rho}(s,a)\iota}{N_h^k(s,a)}} + \sqrt{\frac{32}{N_h^k(s,a)}} + \frac{46\sqrt{SH^4\iota}}{N_h^k(s,a)}$.

**Bound** $\sum_{k=1}^{K} \sum_{h=1}^{H} [(1-\rho)\sqrt{\frac{32\iota}{N_h^k(s_h^k, \overline{\pi}_h^k(s_h^k))}} + \rho\sqrt{\frac{32\iota}{N_h^k(s_h^k, \underline{\pi}_h^k(s_h^k))}}]$   We regroup the summands in a different way. For any policy $\pi$, we have

$$\sum_{k=1}^{K} \sum_{h=1}^{H} \sqrt{\frac{32\iota}{N_h^k(s_h^k, \pi(s_h^k))}} = \sum_{h=1}^{H} \sum_{(s,a)\in\mathcal{S}\times\mathcal{A}} \sum_{n=1}^{N_h^K(s,a)} \sqrt{\frac{32\iota}{n}} \leq 8H\sqrt{SAK\iota}.$$

(50)

**Bound** $\sum_{k=1}^{K} \sum_{h=1}^{H} [(1-\rho)\frac{46SH^2\iota}{N_h^k(s_h^k, \overline{\pi}_h^k(s_h^k))} + \rho\frac{46SH^2\iota}{N_h^k(s_h^k, \underline{\pi}_h^k(s_h^k))}]$   We regroup the summands in a different way. For any policy $\pi$, we have

$$\sum_{k=1}^{K} \sum_{h=1}^{H} \frac{46\sqrt{SH^4\iota}}{N_h^k(s_h^k, \pi(s_h^k))} = 46\sqrt{SH^4\iota} \sum_{h=1}^{H} \sum_{(s,a)\in\mathcal{S}\times\mathcal{A}} \sum_{n=1}^{N_h^K(s,a)} \frac{1}{n} \leq 46S^{\frac{3}{2}}AH^3\iota^2.$$

(51)

**Bound** $\sum_{k=1}^{K} \sum_{h=1}^{H} \left[ (1-\rho)\sqrt{\frac{8\mathbb{V}_{P_h} C_{h+1}^{\pi^{k*}, \underline{\pi}^k, \rho}(s_h^k, \overline{\pi}_h^k(s_h^k))\iota}{N_h^k(s_h^k, \overline{\pi}_h^k(s_h^k))}} + \rho\sqrt{\frac{8\mathbb{V}_{P_h} C_{h+1}^{\pi^{k*}, \underline{\pi}^k, \rho}(s_h^k, \underline{\pi}_h^k(s_h^k))\iota}{N_h^k(s_h^k, \underline{\pi}_h^k(s_h^k))}} \right]$   By
Cauchy-Schwarz inequality,

$$\sum_{k=1}^{K} \sum_{h=1}^{H} \sqrt{\frac{\mathbb{V}_{P_h} C_{h+1}^{\pi^{k*}, \underline{\pi}^k, \rho}(s_h^k, \overline{\pi}_h^k(s_h^k))\iota}{N_h^k(s_h^k, \overline{\pi}_h^k(s_h^k))}}$$
$$\leq \sqrt{\sum_{k=1}^{K} \sum_{h=1}^{H} \mathbb{V}_{P_h} C_{h+1}^{\pi^{k*}, \underline{\pi}^k, \rho}(s_h^k, \overline{\pi}_h^k(s_h^k)) \cdot \sum_{k=1}^{K} \sum_{h=1}^{H} \frac{\iota}{N_h^k(s_h^k, \overline{\pi}_h^k(s_h^k))}}$$
$$\leq \sqrt{SAH\iota^2 \sum_{k=1}^{K} \sum_{h=1}^{H} \mathbb{V}_{P_h} C_{h+1}^{\pi^{k*}, \underline{\pi}^k, \rho}(s_h^k, \overline{\pi}_h^k(s_h^k))}.$$

(52)

Similarly,

$$\sum_{k=1}^{K} \sum_{h=1}^{H} \sqrt{\frac{\mathbb{V}_{P_h} C_{h+1}^{\pi^{k*}, \underline{\pi}^k, \rho}(s_h^k, \underline{\pi}_h^k(s_h^k))\iota}{N_h^k(s_h^k, \underline{\pi}_h^k(s_h^k))}}$$
$$\leq \sqrt{SAH\iota^2 \sum_{k=1}^{K} \sum_{h=1}^{H} \mathbb{V}_{P_h} C_{h+1}^{\pi^{k*}, \underline{\pi}^k, \rho}(s_h^k, \underline{\pi}_h^k(s_h^k))}.$$

(53)

By $(1-\rho)a^2 + \rho b^2 \geq ((1-\rho)a + \rho b)^2$,

$$(1-\rho)\sqrt{\sum_{k=1}^{K} \sum_{h=1}^{H} \mathbb{V}_{P_h} C_{h+1}^{\pi^{k*}, \underline{\pi}^k, \rho}(s_h^k, \overline{\pi}_h^k(s_h^k))} + \rho\sqrt{\sum_{k=1}^{K} \sum_{h=1}^{H} \mathbb{V}_{P_h} C_{h+1}^{\pi^{k*}, \underline{\pi}^k, \rho}(s_h^k, \underline{\pi}_h^k(s_h^k))}$$
$$\leq \sqrt{\sum_{k=1}^{K} \sum_{h=1}^{H} [(1-\rho)\mathbb{V}_{P_h} C_{h+1}^{\pi^{k*}, \underline{\pi}^k, \rho}(s_h^k, \overline{\pi}_h^k(s_h^k)) + \rho\mathbb{V}_{P_h} C_{h+1}^{\pi^{k*}, \underline{\pi}^k, \rho}(s_h^k, \underline{\pi}_h^k(s_h^k))]}.$$

(54)

Now we bound the total variance. Let $\mathbb{D}_{\widetilde{\pi}_h^k}P_h(s'|s) = (1-\rho)P_h(s'|s,\overline{\pi}_h^k(s)) + \rho P_h(s'|s,\underline{\pi}_h^k(s))$,

$$[\mathbb{D}_{\widetilde{\pi}_h^k}P_hV_{h+1}](s) = \sum_{s'}[(1-\rho)P_h(s'|s,\overline{\pi}_h^k(s)) + \rho P_h(s'|s,\underline{\pi}_h^k(s))]V_{h+1}(s'), \tag{55}$$

and

$$\mathbb{V}_{[\mathbb{D}_{\widetilde{\pi}_h^k}P_h]}V_{h+1}(s) = \sum_{s'}[(1-\rho)P_h(s'|s,\overline{\pi}_h^k(s)) + \rho P_h(s'|s,\underline{\pi}_h^k(s))][V_{h+1}(s')]^2$$
$$- [\sum_{s'}\left((1-\rho)P_h(s'|s,\overline{\pi}_h^k(s)) + \rho P_h(s'|s,\underline{\pi}_h^k(s))\right)V_{h+1}(s')]^2. \tag{56}$$

We have that

$$\mathbb{V}_{[\mathbb{D}_{\widetilde{\pi}_h^k}P_h]}C_{h+1}^{\pi^{k*},\underline{\pi}^k,\rho}(s_h^k)$$

$$= \sum_{s'}[(1-\rho)P_h(s'|s_h^k,\overline{\pi}_h^k(s_h^k)) + \rho P_h(s'|s_h^k,\underline{\pi}_h^k(s_h^k))][C_{h+1}^{\pi^{k*},\underline{\pi}^k,\rho}(s')]^2$$

$$- [\sum_{s'}\left((1-\rho)P_h(s'|s_h^k,\overline{\pi}_h^k(s_h^k)) + \rho P_h(s'|s_h^k,\underline{\pi}_h^k(s_h^k))\right)C_{h+1}^{\pi^{k*},\underline{\pi}^k,\rho}(s')]^2$$

$$\geq (1-\rho)\mathbb{V}_{P_h}C_{h+1}^{\pi^{k*},\underline{\pi}^k,\rho}(s_h^k,\overline{\pi}_h^k(s_h^k)) + \rho\mathbb{V}_{P_h}C_{h+1}^{\pi^{k*},\underline{\pi}^k,\rho}(s_h^k,\underline{\pi}_h^k(s_h^k)) \tag{57}$$

$$+ (1-\rho)[P_hC_{h+1}^{\pi^{k*},\underline{\pi}^k,\rho}(s_h^k,\overline{\pi}_h^k(s_h^k))]^2 + \rho P_h[C_{h+1}^{\pi^{k*},\underline{\pi}^k,\rho}(s_h^k,\underline{\pi}_h^k(s_h^k))]^2$$

$$- [\sum_{s'}(1-\rho)P_h(s'|s_h^k,\overline{\pi}_h^k(s_h^k))C_{h+1}^{\pi^{k*},\underline{\pi}^k,\rho}(s') + \rho P_h(s'|s_h^k,\underline{\pi}_h^k(s_h^k))C_{h+1}^{\pi^{k*},\underline{\pi}^k,\rho}(s')]^2$$

$$\geq (1-\rho)\mathbb{V}_{P_h}C_{h+1}^{\pi^{k*},\underline{\pi}^k,\rho}(s_h^k,\overline{\pi}_h^k(s_h^k)) + \rho\mathbb{V}_{P_h}C_{h+1}^{\pi^{k*},\underline{\pi}^k,\rho}(s_h^k,\underline{\pi}_h^k(s_h^k)),$$

where the last inequality is due to $(1-\rho)a^2 + \rho b^2 \geq ((1-\rho)a + \rho b)^2$.

With probability $1-2\delta$, we also have that

$$\sum_{k=1}^K\sum_{h=1}^H\mathbb{V}_{[\mathbb{D}_{\widetilde{\pi}_h^k}P_h]}C_{h+1}^{\pi^{k*},\underline{\pi}^k,\rho}(s_h^k)$$

$$= \sum_{k=1}^K\sum_{h=1}^H\left([\mathbb{D}_{\widetilde{\pi}_h^k}P_h(C_{h+1}^{\pi^{k*},\underline{\pi}^k,\rho})^2](s_h^k) - \left([\mathbb{D}_{\widetilde{\pi}_h^k}P_hC_{h+1}^{\pi^{k*},\underline{\pi}^k,\rho}](s_h^k)\right)^2\right)$$

$$= \sum_{k=1}^K\sum_{h=1}^H\left([\mathbb{D}_{\widetilde{\pi}_h^k}P_h(C_{h+1}^{\pi^{k*},\underline{\pi}^k,\rho})^2](s_h^k) - \left(C_{h+1}^{\pi^{k*},\underline{\pi}^k,\rho}(s_{h+1}^k)\right)^2\right)$$

$$+ \sum_{k=1}^K\sum_{h=1}^H\left(\left(C_{h+1}^{\pi^{k*},\underline{\pi}^k,\rho}(s_{h+1}^k)\right)^2 - \left([\mathbb{D}_{\widetilde{\pi}_h^k}P_hC_{h+1}^{\pi^{k*},\underline{\pi}^k,\rho}](s_h^k)\right)^2\right)$$

$$\leq H^2\sqrt{2HK\iota} + \sum_{k=1}^K\sum_{h=1}^H\left((C_h^{\pi^{k*},\underline{\pi}^k,\rho}(s_h^k))^2 - \left([\mathbb{D}_{\widetilde{\pi}_h^k}P_hC_{h+1}^{\pi^{k*},\underline{\pi}^k,\rho}](s_h^k)\right)^2\right) - \sum_{k=1}^K(C_1^{\pi^{k*},\underline{\pi}^k,\rho}(s_1^k))^2$$

$$\leq H^2\sqrt{2HK\iota} + 2H\sum_{k=1}^K\sum_{h=1}^H|C_h^{\pi^{k*},\underline{\pi}^k,\rho}(s_h^k) - \mathbb{D}_{\widetilde{\pi}_h^k}P_hC_{h+1}^{\pi^{k*},\underline{\pi}^k,\rho}(s_h^k)|$$

$$\leq H^2\sqrt{2HK\iota} + 2H\sum_{k=1}^K\left(C_1^{\pi^{k*},\underline{\pi}^k,\rho}(s_1^k) + \sum_{h=1}^H\left(C_{h+1}^{\pi^{k*},\underline{\pi}^k,\rho}(s_{h+1}^k) - \mathbb{D}_{\widetilde{\pi}_h^k}P_hC_{h+1}^{\pi^{k*},\underline{\pi}^k,\rho}(s_h^k,a_h^k)\right)\right)$$

$$\leq H^2\sqrt{2HK\iota} + 2H^2K + 2H^2\sqrt{2HK\iota}$$

$$\leq 3H^2K + 9H^3\iota/2, \tag{58}$$

where the first inequality holds with probability $1-\delta$ by Azuma-Hoeffding inequality, the second inequality is due to the bound of V-values, the third inequality is due to Lemma 2 so that

$C_h^{\pi^{k*},\underline{\pi}^k,\rho}(s_h^k) \geq \mathbb{D}_{\widetilde{\pi}_h^k} D_h^{\pi^{k*},\underline{\pi}^k,\rho}(s_h^k) \geq \mathbb{D}_{\widetilde{\pi}_h^k} P_h C_{h+1}^{\pi^{k*},\underline{\pi}^k,\rho}(s_h^k)$, the fourth inequality holds with probability $1 - \delta$ by Azuma-Hoeffding inequality, and the last inequality holds with $2ab \leq a^2 + b^2$.

In summary, with probability at least $1-\delta$, we have $\sum_{k=1}^{K} \sum_{h=1}^{H} \mathbb{V}_{P_h} V_{h+1}^{\overline{\pi}^k}(s_h^k, a_h^k) \leq (H^2 K + H^3 \iota)$.

In summary, $\sum_{k=1}^{K} \sum_{h=1}^{H} \mathbb{D}_{\widetilde{\pi}_h^k} \Theta_h^k(s_h^k) \leq 8\sqrt{SAH^2 K\iota} + 46S^{\frac{3}{2}}AH^3\iota^2 + \sqrt{24SAH^3 K\iota^2 + 36SAH^5\iota^2} \leq 8\sqrt{SAH^2 K\iota} + 46S^{\frac{3}{2}}AH^3\iota^2 + \sqrt{24SAH^3 K\iota} + 6\sqrt{SAH^5\iota}$.

## D  MODEL-FREE METHOD

In this section, we develop a model-free algorithm and analyze its theoretical guarantee. We present the proposed Action Robust Q-learning with UCB-Hoeffding (AR-UCBH) algorithm show in Algorithm 2. Here, we highlight the main idea of Algorithm 2. Algorithm 2 follows the same idea of Algorithm 1, which trains the agent in a clean (simulation) environment and learns a policy that performs well when applied to a perturbed environment with probabilistic policy execution uncertainty. To simulate the action perturbation, Algorithm 2 chooses an adversarial action with probability $\rho$. To learn the agent's optimal policy and the corresponding adversarial policy, Algorithm 2 computes an optimistic estimate $\overline{Q}$ of $Q^*$ and a pessimistic estimate $\underline{Q}$ of $Q^{\overline{\pi}^k}$. Algorithm 2 uses the optimistic estimates to explore the possible optimal policy $\overline{\pi}$ and uses the pessimistic estimates to explore the possible adversarial policy $\underline{\pi}$. The difference is that Algorithm 2 use a model-free method to update $Q$ and $V$ values.

---

**Algorithm 2:** Action Robust Q-learning with UCB-Hoeffding (AR-UCBH)

---

1: Set $\alpha_t = \frac{H+1}{H+t}$. Initialize $\overline{V}_h(s) = H - h + 1, \overline{Q}_h(s,a) = H - h + 1, \underline{V}_h(s) = 0,$
   $\underline{Q}_h(s,a) = 0, \hat{r}_h(s,a), N_h(s,a) = 0$ for any state $s \in \mathcal{S}$, any action $a \in \mathcal{A}$ and any step
   $h \in [H]$. $\overline{V}_{H+1}(s) = \underline{V}_{H+1}(s) = 0$ and $\overline{Q}_{H+1}(s,a) = \underline{Q}_{H+1}(s,a) = 0$ for all $s$ and $a$.
   $\Delta = H$. Initial policy $\overline{\pi}_h^1(a|s)$ and $\underline{\pi}_h^1(a|s) = 1/A$ for any state $s$, action $a$ and any step
   $h \in [H]$.
2: **for** episode $k = 1, 2, \ldots, K$ **do**
3:     **for** step $h = 1, 2, \ldots, H$ **do**
4:         Observe $s_h^k$.
5:         Set $\overline{a}_h^k = \arg\max_a \overline{Q}_h(s_h^k, a)$ , $\underline{a}_h^k = \arg\min_a \underline{Q}_h(s_h^k, a), \widetilde{\pi}_h^k(\overline{a}_h^k|s_h^k) = 1 - \rho$ and
           $\widetilde{\pi}_h^k(\underline{a}_h^k|s_h^k) = \rho$.
6:         Take action $a_h^k \sim \widetilde{\pi}_h^k(\cdot|s_h^k)$.
7:         Receive reward $r_h^k$ and observe $s_{h+1}^k$.
8:         Set $t = N_h(s_h^k, a_h^k) \leftarrow N_h(s_h^k, a_h^k) + 1; b_t = \sqrt{H^3 \iota/t}$.
9:         $\overline{Q}_h(s_h^k, a_h^k) \leftarrow (1 - \alpha_t)\overline{Q}_h(s_h^k, a_h^k) + \alpha_t(r_h^k + \overline{V}_{h+1}(s_{h+1}^k) + b_t),$
10:        $\underline{Q}_h(s_h^k, a_h^k) \leftarrow (1 - \alpha_t)\underline{Q}_h(s_h^k, a_h^k) + \alpha_t(r_h^k + \underline{V}_{h+1}(s_{h+1}^k) - b_t).$
11:        Set $\overline{\pi}_h^{k+1}(s_h^k) = \arg\max_a \overline{Q}_h(s_h^k, a), \underline{\pi}_h^{k+1}(s_h^k) = \arg\min_a \underline{Q}_h(s_h^{k+1}, a).$
12:        $\overline{V}_h(s_h^k) \leftarrow \min\{\overline{V}_h(s_h^k), (1-\rho)\overline{Q}_h(s_h^k, \overline{\pi}_h^{k+1}(s_h^k)) + \rho\overline{Q}_h(s_h^k, \underline{\pi}_h^{k+1}(s_h^k))\}.$
13:        $\underline{V}_h(s_h^k) \leftarrow \max\{\underline{V}_h(s_h^k), (1-\rho)\underline{Q}_h(s_h^k, \overline{\pi}_h^{k+1}(s_h^k)) + \rho\underline{Q}_h(s_h^k, \underline{\pi}_h^{k+1}(s_h^k))\}.$
14:        **if** $\underline{V}_h(s_h^k) > (1-\rho)\underline{Q}_h(s_h^k, \overline{\pi}_h^{k+1}(s_h^k)) + \rho\underline{Q}_h(s_h^k, \underline{\pi}_h^{k+1}(s_h^k))$ **then**
15:            $\overline{\pi}_h^{k+1} = \overline{\pi}_h^k.$
16:        **end if**
17:     **end for**
18:     **Output** policy $\overline{\pi}^{k+1}$ with certificates $\mathcal{I}_{k+1} = [\underline{V}_1(s_1^k), \overline{V}_1(s_1^k)]$ and $\epsilon_{k+1} = |\mathcal{I}_{k+1}|$.
19: **end for**
20: **return** $\overline{\pi}^{k+1}$

---

Here, we highlight the challenges of the model-free planning compared with the model-based planing. In the model-based planning, we performs value iteration and the $Q$ values, $V$ values, agent policy $\overline{\pi}$ and adversarial policy $\underline{\pi}$ are updated on all $(s, a)$. However, in the model-free method, the $Q$ values, $V$ values are updated only on $(s_h^k, a_h^k)$ which are the samples on the trajectories. Compared with

the model-based planning, the model-free planning is slower and less stable. We need to update the output policy carefully. In line 14-16, Algorithm 2 does not update the output policy when the lower bound on the value function of the new policy does not improve. By this, the output policies are stably updated.

We provide the regret and sample complexity bounds of Algorithm 2 in the following:

**Theorem 2** *For any* $\delta \in (0, 1]$, *letting* $\iota = \log(2SABHK/\delta)$, *then with probability at least* $1 - \delta$, *Algorithm 2 achieves:*

- $V_1^*(s_1) - V_1^{\pi^{out}}(s_1) \leq \epsilon$, *if the number of episodes* $K \geq \Omega(SAH^5\iota/\epsilon^2 + SAH^2/\epsilon)$.

- $Regret(K) = \sum_{k=1}^K (V_1^*(s_1^k) - V_1^{\overline{\pi}^k}(s_1^k)) \leq \mathcal{O}(\sqrt{SAH^5K\iota} + SAH^2)$.

The detailed proof is provided in Appendix E

## E   PROOF FOR MODEL-FREE ALGORITHM

In this section, we prove Theorem 2. Recall that we use $\overline{Q}_h^k, \overline{V}_h^k, \underline{Q}_h^k, \underline{V}_h^k$ and $N_h^k$ to denote the values of $\overline{Q}_h, \overline{V}_h, \underline{Q}_h, \underline{V}_h$ and $\max\{N_h, 1\}$ at the beginning of the $k$-th episode.

**Property of Learning Rate** $\alpha_t$    We refer the readers to the setting of the learning rate $\alpha_t := \frac{H+1}{H+t}$ and the Lemma 4.1 in (Jin et al., 2018). For notational convenience, define $\alpha_t^0 := \prod_{j=1}^t (1 - \alpha_t)$ and $\alpha_t^i := \alpha_i \prod_{j=i+1}^t (1 - \alpha_t)$. Here, we introduce some useful properties of $\alpha_t^i$ which were proved in (Jin et al., 2018):
(1) $\sum_{i=1}^t \alpha_t^i = 1$ and $\alpha_t^0 = 0$ for $t \geq 1$;
(2) $\sum_{i=1}^t \alpha_t^i = 0$ and $\alpha_t^0 = 1$ for $t = 0$;
(3) $\frac{1}{\sqrt{t}} \leq \sum_{i=1}^t \frac{\alpha_t^i}{\sqrt{t}} \leq \frac{2}{\sqrt{t}}$ for every $t \geq 1$;
(4) $\sum_{i=1}^t (\alpha_t^i)^2 \leq \frac{2H}{t}$ for every $t \geq 1$;
(5) $\sum_{t=i}^\infty \alpha_t^i \leq (1 + \frac{1}{H})$ for every $i \geq 1$.

**Recursion on** $Q$    As shown in (Jin et al., 2018), at any $(s, a, h, k) \in \mathcal{S} \times \mathcal{A} \times [H] \times [K]$, let $t = N_h^k(s, a)$ and suppose $(s, a)$ was previously taken by the agent at step $h$ of episodes $k_1, k_2, \ldots, k_t < k$. By the update equations in Algorithm 2 and the definition of $\alpha_t^i$, we have

$$\overline{Q}_h^k(s, a) = \alpha_t^0(H - h + 1) + \sum_{i=1}^t \alpha_t^i \left( r_h^{k_i} + \overline{V}_{h+1}^{k_i}(s_{h+1}^{k_i}) + b_i \right);$$

$$\underline{Q}_h^k(s, a) = \sum_{i=1}^t \alpha_t^i \left( r_h^{k_i} + \underline{V}_{h+1}^{k_i}(s_{h+1}^{k_i}) - b_i \right).$$

(59)

Thus,

$$
\begin{aligned}
(\overline{Q}_h^k - Q_h^*)(s, a) =& \alpha_t^0(H - h + 1) + \sum_{i=1}^t \alpha_t^i \left( r_h^{k_i} + \overline{V}_{h+1}^{k_i}(s_{h+1}^{k_i}) + b_i \right) \\
& - \left( \alpha_t^0 Q_h^*(s, a) + \sum_{i=1}^t \alpha_t^i \left( R_h(s, a) + P_h V_{h+1}^*(s, a) \right) \right) \\
=& \alpha_t^0(H - h + 1 - Q_h^*(s, a)) + \sum_{i=1}^t \alpha_t^i \left( (\overline{V}_{h+1}^{k_i} - V_{h+1}^*)(s_{h+1}^{k_i}) \right) \\
& + \sum_{i=1}^t \alpha_t^i \left( (r_h^{k_i} - R_h(s, a)) + V_{h+1}^*(s_{h+1}^{k_i}) - P_h V_{h+1}^*(s, a) + b_i \right),
\end{aligned}
$$

(60)

and similarly

$$
\begin{aligned}
(\underline{Q}_h^k - Q_h^{\overline{\pi}^k})(s,a) =& \sum_{i=1}^t \alpha_t^i \left( r_h^{k_i} + \underline{V}_{h+1}^{k_i}(s_{h+1}^{k_i}) - b_i \right) \\
& - \left( \alpha_t^0 Q_h^{\overline{\pi}^k}(s,a) + \sum_{i=1}^t \alpha_t^i \left( R_h(s,a) + P_h V_{h+1}^{\overline{\pi}^k}(s,a) \right) \right) \\
=& -\alpha_t^0 Q_h^{\overline{\pi}^k}(s,a) + \sum_{i=1}^t \alpha_t^i \left( [P_h(\underline{V}_{h+1}^{k_i} - V_{h+1}^{\overline{\pi}^k})](s,a) \right) \\
& + \sum_{i=1}^t \alpha_t^i \left( (r_h^{k_i} - R_h(s,a)) + \underline{V}_{h+1}^{k_i}(s_{h+1}^{k_i}) - P_h \underline{V}_{h+1}^{k_i}(s,a) - b_i \right).
\end{aligned}
\tag{61}
$$

In addition, for any $k' \leq k$, let $t' = N_h^{k'}(s,a)$. Thus, $(s,a)$ was previously taken by the agent at step $h$ of episodes $k_1, k_2, \ldots, k_{t'} < k'$. We have

$$
\begin{aligned}
(\underline{Q}_h^{k'} - Q_h^{\overline{\pi}^k})(s,a) =& -\alpha_t^0 Q_h^{\overline{\pi}^k}(s,a) + \sum_{i=1}^{t'} \alpha_{t'}^i \left( [P_h(\underline{V}_{h+1}^{k_i} - V_{h+1}^{\overline{\pi}^k})](s,a) \right) \\
& + \sum_{i=1}^{t'} \alpha_{t'}^i \left( (r_h^{k_i} - R_h(s,a)) + \underline{V}_{h+1}^{k_i}(s_{h+1}^{k_i}) - P_h \underline{V}_{h+1}^{k_i}(s,a) - b_i \right).
\end{aligned}
\tag{62}
$$

**Confidence Bounds**  By the Azuma-Hoeffding inequality, with probability $1 - \delta$, we have that for all $s$, $a$, $h$ and $t \leq K$,

$$
\left| \sum_{i=1}^t \alpha_t^i \left( (r_h^{k_i} - R_h(s,a)) + \underline{V}_{h+1}^{k_i}(s_{h+1}^{k_i}) - P_h \underline{V}_{h+1}^{k_i}(s,a) \right) \right| \leq H \sqrt{\sum_{i=1}^t (\alpha_t^i)^2 \iota/2} \leq \sqrt{H^3 \iota/t}.
\tag{63}
$$

At the same time, with probability $1 - \delta$, we have that for all $s$, $a$, $h$ and $t \leq K$,

$$
\left| \sum_{i=1}^t \alpha_t^i \left( (r_h^{k_i} - R_h(s,a)) + V_{h+1}^*(s_{h+1}^{k_i}) - P_h V_{h+1}^*(s,a) \right) \right| \leq \sqrt{H^3 \iota/t}.
\tag{64}
$$

In addition, we have $\sqrt{H^3 \iota/t} \leq \sum_{i=1}^t \alpha_t^i b_i \leq 2\sqrt{H^3 \iota/t}$.

**Monotonicity**  Now we prove that $\overline{V}_h^k(s) \geq V_h^*(s) \geq V_h^{\overline{\pi}^k}(s) \geq \underline{V}_h^k(s)$ and $\overline{Q}_h^k(s,a) \geq Q_h^*(s,a) \geq Q_h^{\overline{\pi}^k}(s,a) \geq \underline{Q}_h^k(s,a)$ for all $(s,a,h,k) \in S \times A \times [H] \times [K]$.

At step $H + 1$, we have $\overline{V}_{H+1}^k(s) = V_{H+1}^*(s) = V_{H+1}^{\overline{\pi}^k}(s) = \underline{V}_{H+1}^k(s) = 0$ and $\overline{Q}_{H+1}^k(s,a) = Q_{H+1}^*(s,a) = Q_{H+1}^{\overline{\pi}^k}(s,a) = \underline{Q}_{H+1}^k(s,a) = 0$ for all $(s,a,k) \in S \times A \times [K]$.

Consider any step $h \in [H]$ in any episode $k \in [K]$, and suppose that the monotonicity is satisfied for all previous episodes as well as all steps $h' \geq h + 1$ in the current episode, which is

$$
\begin{aligned}
& \overline{V}_{h'}^{k'}(s) \geq V_{h'}^*(s) \geq V_{h'}^{\overline{\pi}^{k'}}(s) \geq \underline{V}_{h'}^{k'}(s) \; \forall (k',h',s) \in [k-1] \times [H+1] \times S, \\
& \overline{Q}_{h'}^{k'}(s,a) \geq Q_{h'}^*(s,a) \geq Q_{h'}^{\overline{\pi}^{k'}}(s,a) \geq \underline{Q}_{h'}^{k'}(s,a) \; \forall (k',h',s,a) \in [k-1] \times [H+1] \times S \times A, \\
& \overline{V}_{h'}^k(s) \geq V_{h'}^*(s) \geq V_{h'}^{\overline{\pi}^k}(s) \geq \underline{V}_{h'}^k(s) \; \forall h' \geq h+1 \text{ and } s \in S, \\
& \overline{Q}_{h'}^k(s,a) \geq Q_{h'}^*(s,a) \geq Q_{h'}^{\overline{\pi}^k}(s,a) \geq \underline{Q}_{h'}^k(s,a) \; \forall h' \geq h+1 \text{ and } (s,a) \in S \times A.
\end{aligned}
\tag{65}
$$

We first show the monotonicity of $Q$ values. We have

$$
(\overline{Q}_h^k - Q_h^*)(s,a) \geq \alpha_t^0(H - h + 1 - Q_h^*(s,a)) + \sum_{i=1}^t \alpha_t^i \left( (\overline{V}_{h+1}^{k_i} - V_{h+1}^*)(s_{h+1}^{k_i}) \right) \geq 0,
\tag{66}
$$

and, by to the update rule of $\underline{V}$ values (line 13) in Algorithm 2,

$$
\begin{aligned}
(\underline{Q}_h^k - Q_h^{\overline{\pi}^k})(s,a) &\leq -\alpha_t^0 Q_h^{\overline{\pi}^k}(s,a) + \sum_{i=1}^t \alpha_t^i \left([P_h(\underline{V}_{h+1}^{k_i} - V_{h+1}^{\overline{\pi}^k})](s,a)\right) \\
&\leq -\alpha_t^0 Q_h^{\overline{\pi}^k}(s,a) + \sum_{i=1}^t \alpha_t^i \left([P_h(\underline{V}_{h+1}^k - V_{h+1}^{\overline{\pi}^k})](s,a)\right) \leq 0.
\end{aligned}
\tag{67}
$$

In addition, for any $k' \leq k$,

$$
\begin{aligned}
(\underline{Q}_h^{k'} - Q_h^{\overline{\pi}^k})(s,a) &\leq -\alpha_t^0 Q_h^{\overline{\pi}^k}(s,a) + \sum_{i=1}^{t'} \alpha_{t'}^i \left([P_h(\underline{V}_{h+1}^{k_i} - V_{h+1}^{\overline{\pi}^k})](s,a)\right) \\
&\leq -\alpha_t^0 Q_h^{\overline{\pi}^k}(s,a) + \sum_{i=1}^{t'} \alpha_{t'}^i \left([P_h(\underline{V}_{h+1}^k - V_{h+1}^{\overline{\pi}^k})](s,a)\right) \leq 0.
\end{aligned}
\tag{68}
$$

Then, we show the monotonicity of $V$ values. We have that

$$
\begin{aligned}
&(1-\rho)\max_a \overline{Q}_h^k(s,a) + \rho\overline{Q}_h^k(s, \arg\min_a \underline{Q}_h^k(s,a)) \\
\geq &(1-\rho)\max_a \overline{Q}_h^k(s,a) + \rho Q_h^*(s, \arg\min_a \underline{Q}_h^k(s,a)) \\
\geq &(1-\rho)\overline{Q}_h^k(s, \pi_h^*(s)) + \rho\min_{a\in\mathcal{A}} Q_h^*(s,a) \\
\geq &(1-\rho)Q_h^*(s, \pi_h^*(s)) + \rho\min_{a\in\mathcal{A}} Q_h^*(s,a) = V_h^*(s).
\end{aligned}
\tag{69}
$$

By the update rule of $\overline{V}$ values (line 12) in Algorithm 2,

$$
\overline{V}_h^k(s) = \min\{\overline{V}_h^{k-1}(s), (1-\rho)\max_a \overline{Q}_h^k(s,a) + \rho\overline{Q}_h^k(s, \arg\min_a \underline{Q}_h^k(s,a))\} \geq V_h^*(s).
\tag{70}
$$

Here, we need use the update rule of policy $\underline{\pi}$ (line 11-16) in Algorithm 2. Define $\tau(k,h,s) := \max\{k' : k' < k \text{ and } \underline{V}_h^{k'+1}(s) = (1-\rho)\underline{Q}_h^{k'+1}(s, \arg\max_a \overline{Q}_h^{k'+1}(s,a)) + \rho\min_a \underline{Q}_h^{k'+1}(s,a)\}$, which denotes the last episode (before the beginning of the episode $k$), in which the $\overline{\pi}$ and $\underline{V}$ was updated at $(h,s)$. For notational simplicity, we use $\tau$ to denote $\tau(k,h,s)$ here. After the end of episode $\tau$ and before the beginning of the episode $k$, the agent policy $\overline{\pi}$ was not updated and $\underline{V}$ was not updated at $(h,s)$, i.e. $\underline{V}_h^k(s) = \underline{V}_h^{\tau+1}(s) = (1-\rho)\underline{Q}_h^{\tau+1}(s, \overline{\pi}_h^{\tau+1}(s)) + \rho\min_a \underline{Q}_h^{\tau+1}(s,a)$ and $\overline{\pi}_h^k(s) = \overline{\pi}_h^{\tau+1}(s) = \arg\max_a \overline{Q}_h^{\tau+1}(s,a)$. Thus,

$$
\begin{aligned}
\underline{V}_h^k(s) &= (1-\rho)\underline{Q}_h^{\tau+1}(s, \overline{\pi}_h^{\tau+1}(s)) + \rho\min_a \underline{Q}_h^{\tau+1}(s,a) \\
&\leq (1-\rho)Q_h^{\overline{\pi}^k}(s, \overline{\pi}_h^{\tau+1}(s)) + \rho\min_a \underline{Q}_h^{\tau+1}(s,a) \\
&\leq (1-\rho)Q_h^{\overline{\pi}^k}(s, \overline{\pi}_h^k(s)) + \rho\underline{Q}_h^{\tau+1}(s, \arg\min_{a\in\mathcal{A}} Q_h^{\overline{\pi}^k}(s,a)) \\
&\leq (1-\rho)Q_h^{\overline{\pi}^k}(s, \overline{\pi}_h^k(s)) + \rho\min_{a\in\mathcal{A}} Q_h^{\overline{\pi}^k}(s,a) = V_h^{\overline{\pi}^k}(s).
\end{aligned}
\tag{71}
$$

By induction from $h = H + 1$ to 1 and $k = 1$ to $K$, we can conclude that $\overline{V}_h^k(s) \geq V_h^*(s) \geq V_h^{\overline{\pi}^k}(s) \geq \underline{V}_h^k(s)$ and $\overline{Q}_h^k(s,a) \geq Q_h^*(s,a) \geq Q_h^{\overline{\pi}^k}(s,a) \geq \underline{Q}_h^k(s,a)$ for all $(s,a,h,k) \in S \times A \times [H] \times [K]$.

**Regret Analysis** According to the monotonicity, the regret can be bounded by

$$
Regret(K) := \sum_{k=1}^K (V_1^*(s_1^k) - V_1^{\overline{\pi}^k}(s_1^k)) \leq \sum_{k=1}^K (\overline{V}_1^k(s_1^k) - \underline{V}_1^k(s_1^k)).
\tag{72}
$$

By the update rules in Algorithm 2, we have

$$\overline{V}_h^k(s_h^k) - \underline{V}_h^k(s_h^k)$$

$$\leq (1-\rho)\overline{Q}_h^k(s_h^k, \arg\max_a \overline{Q}_h^k(s_h^k, a)) + \rho\overline{Q}_h^k(s_h^k, \arg\min_a \underline{Q}_h^k(s_h^k, a))$$

$$- (1-\rho)\underline{Q}_h^k(s_h^k, \arg\max_a \overline{Q}_h^k(s_h^k, a)) + \rho\underline{Q}_h^k(s_h^k, \arg\min_a \underline{Q}_h^k(s_h^k, a)) \tag{73}$$

$$= (1-\rho)[\overline{Q}_h^k - \underline{Q}_h^k](s_h^k, \overline{a}_h^k) + \rho[\overline{Q}_h^k - \underline{Q}_h^k](s_h^k, \underline{a}_h^k)$$

$$= [\overline{Q}_h^k - \underline{Q}_h^k](s_h^k, a_h^k) + [\mathbb{D}_{\widetilde{\pi}_h^k}(\overline{Q}_h^k - \underline{Q}_h^k)](s_h^k) - [\overline{Q}_h^k - \underline{Q}_h^k](s_h^k, a_h^k).$$

Set $n_h^k = N_h^k(s_h^k, a_h^k)$ and where $k_i(s_h^k, a_h^k)$ is the episode in which $(s_h^k, a_h^k)$ was taken at step $h$ for the $i$-th time. For notational simplicity, we set $\phi_h^k = \overline{V}_h^k(s_h^k) - \underline{V}_h^k(s_h^k)$ and $\xi_h^k = [\mathbb{D}_{\widetilde{\pi}_h^k}(\overline{Q}_h^k - \underline{Q}_h^k)](s_h^k) - [\overline{Q}_h^k - \underline{Q}_h^k](s_h^k, a_h^k)$. According to the update rules,

$$\phi_h^k = \overline{V}_h^k(s_h^k) - \underline{V}_h^k(s_h^k)$$

$$\leq \alpha_{n_h^k}^0(H - h + 1) + \sum_{i=1}^{n_h^k} \alpha_{n_h^k}^i \left( \overline{V}_{h+1}^{k_i(s_h^k, a_h^k)}(s_{h+1}^{k_i(s_h^k, a_h^k)}) - \underline{V}_{h+1}^{k_i(s_h^k, a_h^k)}(s_{h+1}^{k_i(s_h^k, a_h^k)}) + 2b_i \right)$$

$$+ [\mathbb{D}_{\widetilde{\pi}_h^k}(\overline{Q}_h^k - \underline{Q}_h^k)](s_h^k) - [\overline{Q}_h^k - \underline{Q}_h^k](s_h^k, a_h^k) \tag{74}$$

$$= \alpha_{n_h^k}^0(H - h + 1) + \sum_{i=1}^{n_h^k} \alpha_{n_h^k}^i(\phi_{h+1}^{k_i(s_h^k, a_h^k)} + 2b_i) + \xi_h^k$$

$$\leq \alpha_{n_h^k}^0(H - h + 1) + \sum_{i=1}^{n_h^k} \alpha_{n_h^k}^i \phi_{h+1}^{k_i(s_h^k, a_h^k)} + \xi_h^k + 4\sqrt{H^3\iota/n_h^k}.$$

We add $\overline{V}_h^k(s_h^k) - \underline{V}_h^k(s_h^k)$ over $k$ and regroup the summands in a different way. Note that for any episode $k$, the term $\sum_{i=1}^{n_h^k} \alpha_{n_h^k}^i \phi_{h+1}^{k_i(s_h^k, a_h^k)}$ takes all the prior episodes $k_i < k$ where $(s_h^k, a_h^k)$ was taken into account. In other words, for any episode $k'$, the term $\phi_{h+1}^{k'}$ appears in the summands at all posterior episodes $k > k'$ where $(s_h^{k'}, a_h^{k'})$ was taken. The first time it appears we have $n_h^k = n_h^{k'} + 1$, and the second time it appears we have $n_h^k = n_h^{k'} + 2$, and so on. Thus, we have

$$\sum_{k=1}^K (\overline{V}_h^k(s_h^k) - \underline{V}_h^k(s_h^k))$$

$$\leq \sum_{k=1}^K \alpha_{n_h^k}^0(H - h + 1) + \sum_{k=1}^K \sum_{i=1}^{n_h^k} \alpha_{n_h^k}^i \phi_{h+1}^{k_i(s_h^k, a_h^k)} + \sum_{k=1}^K \xi_h^k + \sum_{k=1}^K 4\sqrt{H^3\iota/n_h^k}$$

$$= \sum_{k=1}^K \alpha_{n_h^k}^0(H - h + 1) + \sum_{k'=1}^K \phi_{h+1}^{k'} \sum_{t=n_h^{k'}+1}^{n_h^K} \alpha_t^{n_h^{k'}} + \sum_{k=1}^K \xi_h^k + \sum_{k=1}^K 4\sqrt{H^3\iota/n_h^k} \tag{75}$$

$$\leq \sum_{k=1}^K \alpha_{n_h^k}^0(H - h + 1) + (1 + 1/H)\sum_{k=1}^K \phi_{h+1}^k + \sum_{k=1}^K \xi_h^k + \sum_{k=1}^K 4\sqrt{H^3\iota/n_h^k}$$

where the final inequality uses the property $\sum_{t=i}^\infty \alpha_t^i \leq (1 + \frac{1}{H})$ for every $i \geq 1$.

Taking the induction from $h = 1$ to $H$, we have

$$\sum_{k=1}^K (\overline{V}_1^k(s_1^k) - \underline{V}_1^k(s_1^k))$$

$$\leq 3\sum_{h=1}^H \sum_{k=1}^K \alpha_{n_h^k}^0(H - h + 1) + 3\sum_{h=1}^H \sum_{k=1}^K \xi_h^k + \sum_{h=1}^H \sum_{k=1}^K 12\sqrt{H^3\iota/n_h^k} \tag{76}$$

where we use the fact that $(1 + 1/H)^H < 3$ and $\phi_{H+1}^k = 0$ for all $k$.

We bound the three items separately.

(1) We have $\sum_{h=1}^{H} \sum_{k=1}^{K} \alpha_{n_h^k}^0 (H - h + 1) = \sum_{h=1}^{H} \sum_{k=1}^{K} \mathbb{1}[n_h^k = 0](H - h + 1) \leq SAH^2$.

(2) Similar to Lemma 4, by the Azuma-Hoeffding inequality, with probability $1 - \delta$, we have $\sum_{h=1}^{H} \sum_{k=1}^{K} \xi_h^k \leq H\sqrt{2HK\iota}$.

(3) We have $\sum_{h=1}^{H} \sum_{k=1}^{K} 12\sqrt{H^3\iota/n_h^k} = \sum_{h=1}^{H} \sum_{(s,a)} \sum_{n=1}^{N_h^K(s,a)} \sqrt{H^3\iota/n} \leq H\sqrt{2H^3SAK\iota}$.

In summary,

$$Regret(K) = \sum_{k=1}^{K} (V_1^*(s_1^k) - V_1^{\overline{\pi}^k}(s_1^k)) \leq \mathcal{O}(\sqrt{SAH^5K\iota} + SAH^2)$$

and

$$
\begin{aligned}
V_1^*(s_1) - V_1^{\pi^{out}}(s_1) &\leq \overline{V}_1^{K+1}(s_1) - \underline{V}_1^{K+1}(s_1) \\
&= \min_{k \in [K+1]} (\overline{V}_1^k(s_1^k) - \underline{V}_1^k(s_1^k)) \\
&\leq O\left( \frac{\sqrt{SAH^5\iota}}{K} + \frac{SAH^2}{K} \right).
\end{aligned}
\tag{77}
$$

