# OpenReview forum: "Efficient Action Robust Reinforcement Learning with Probabilistic Policy Execution Uncertainty"
_ICLR.cc/2024/Conference — Submitted to ICLR 2024_

### Official Review · Reviewer_N6KC · 2023-10-29

**Soundness:** 3 good
**Presentation:** 3 good
**Contribution:** 1 poor
**Rating:** 6
**Confidence:** 5

**Summary:**

This work provides a learning procedure for the action robust RL problem. This work also gives complementing experiment results.

**Strengths:**

Extension of ORLC algorithm (Dann et al, 2019) to solve action robust RL problem. Extensive experiments have been made to showcase action robustness of ARRLC algorithm. The score reflects the points made in Weaknesses.

**Weaknesses:**

The weaknesses are concerned more likely due to incremental contributions in this work, but I'd rather see this work as a published work rather than a blogpost. I explain this in a few points below.

-  Theorem 1 results in this work are already shown in Proposition 1 of (Tessler et al, 2019). Moreover, even though the latter work does not explicitly define "action robust Bellman equation" and the corresponding optimality equations, they can be inferred from (Eq.(11), Tessler et al, 2019). Although an independent analysis in this work is made for finite-horizon setting whereas (Tessler et al, 2019) is for infinite horizon, the proof ideas share similar characteristics due to the structure of the uncertainty set itself (helping commutability of the expectation and the minimization operations). Thus Theorem 1 proof largely following the proofs in (Iyengar, 2005) for the finite-horizon getting insights from Proposition 1 of (Tessler et al, 2019).

- With the separable form of action robust Bellman equation ($V^*(.) = (1-\rho) \max_{a}Q^*(.,a) + \rho \min_{a}Q^*(.,a)$), ARRLC is an extension of ORLC (Dann et al, 2019) with less technical challenges. The analysis is made finer by replacing $Var(\overline{V})$ with $Var((\overline{V}+\underline{V})/2)$ which is straightforward from Eq.31 due to monotonicity and induction.

I am open to discussions with the authors and reviewers to increase/maintain (already reflects the positive impact) my score. All the best for future decisions!

**Questions:**

-n/a-

---

> ### Author Response · Authors · 2023-11-18
> **Response to Reviewer N6KC**
>
> **To weakness 1.**
>
> We agree with the reviewer that the contributions in Section 4 are incremental. However, the results in Section 4 are not the core contributions of our paper. The main purpose of results in Section 4 is to provide motivation and insights for the design of the proposed two efficient algorithms in later sections. The approaches of Pinto et al. (2017) and Tessler et al. (2019) used an alternating update method that repeatedly solves MDP by fixing either the adversary policy or the agent policy and are sample inefficient. In our proposed algorithm, we use the iteration of action robust Bellman equation to solve the robust problem. Our proposed method can avoid the alternating update between the adversary policy and agent policy. Thus, we need to first introduce the results in Section 4. The proposed action robust Bellman equation in Section 4 tells the reader how to directly solve the robust problem for finite-horizon setting when we know the reward function and transition probabilities, and makes ARRLC more easily understandable.
>
> To reflect the discussion above, we have changed the Theorem 1 to Proposition 1 in the revised version so as to de-emphasize the results in Section 4, changed the description of the main contributions in page 2 and added the discussion about the importance of introducing action robust Bellman optimality equation in the revised paper.
>
> **To weakness 2.**
>
> ARRLC shares a similar form with ORLC, but is not an simple extension of ORLC with less technical challenges. Similar to ORLC, ARRLC considers the upper bound and the lower bound of $Q$-values and $V$-values. However, the trajectory sampling and the update of the value functions in ARRLC are different from those in ORLC, as we considered the action robust Bellman equation. More specifically, ARRLC uses two different actions $\overline{\pi}(s)$ and $\underline{\pi}(s)$ to update the value functions but only samples one of them at each step, while ORLC uses the sampled action $\pi(s)$ to update the value function (refer to Line 12 Algorithm 1 in Dann et al, (2019)). We need to do this because the update of the action robust Bellman equation requires both the maximum and minimum over the actions. Thus, some new technical challenges arises. For example, the empirical variance estimators $Var((\overline{V}+\underline{V})/2)$ sum not only over all episode $k$ and step $h$ but also over two different action $\overline{\pi}(s)$ and $\underline{\pi}(s)$, but only one action, $\overline{\pi}(s)$ or $\underline{\pi}(s)$, is sampled. Thus, the sum of variance is no longer simply the variance of the sum of rewards per episode, and we need new techniques to bound the sum of variance. We deal with this challenge in Appendix page 21-24. Some new techniques are then introduced, e.g., $\text{Var}((\overline{V}+\underline{V})/2)$ is further replaced with $C^{\pi^{k_*,\pi^k,\rho}}$, and different confidence bounds are used in our paper. We added these discussion in Section 5.1 at the top of page 6.

---

> > ### Comment · Reviewer_N6KC · 2023-11-21
> >
> > Thank you for the detailed response. I do think the authors have improved the manuscript compared to the pre-rebuttal stage.
> > I have increased my score and will look forward to decision after reviewers-AC discussion.
> >
> > I'd suggest authors add more detail to the following in the paper. Just providing highlight of the new techniques will improve the presentation further!
> > > Thus, the sum of variance is no longer simply the variance of the sum of rewards per episode, and new techniques are needed.

---

> > > ### Author Response · Authors · 2023-11-21
> > >
> > > Thanks for sharing your opinions! We are delighted that we solved your concerns and you raised your evaluation of our work. We deeply appreciate your insights and suggestions, which help to improve the quality and clarity of our paper. In line with your recommendation, we have added an example of the new techniques in the newest revised version to highlight the new techniques.

---

> ### Author Response · Authors · 2023-11-21
> **A kind remind**
>
> Dear reviewer N6KC,
>
> Thank you for dedicating your time to reviewing our manuscript. Your insights have been invaluable, and we've crafted a detailed response to address the points you raised and polish the paper accordingly. We're confident that our responses and modifications have effectively addressed your concerns. With these changes in mind, we hope you might reconsider and adjust your evaluation of our work.
>
> Should there be any remaining issues, we're more than ready to provide further clarifications. As the rebuttal phase is nearing its deadline, we're looking forward to engaging in a timely discussion. Thank you again for your expertise and guidance!

---

### Official Review · Reviewer_if62 · 2023-10-29

**Soundness:** 3 good
**Presentation:** 2 fair
**Contribution:** 2 fair
**Rating:** 6
**Confidence:** 4

**Summary:**

This paper explores the realm of action-robust reinforcement learning, wherein the policy prescribes an action with a high probability of certainty (1 - $\rho$) while considering an alternative adversarial action with a probability of $\rho$. The authors introduce an action-robust Bellman optimality equation and introduce a novel algorithm named Action Robust Reinforcement Learning with Certificate (ARRLC). The authors further establish the minimax optimal regret and illustrate the superior robust performance of ARRLC when compared to both non-robust algorithms and robust TD methods.

**Strengths:**

* This paper is easy to follow.
* The paper demonstrates technical rigor, with most claims substantiated adequately, and its theoretical guarantee appears to be novel.
* Several toy examples are included to illustrate experimental findings.

**Weaknesses:**

* Given the results in [Tessler, 2019], the first contribution listed on page 2 is not very original (no big difference between episodic setting and infinite discounted horizon setting).
* All regret analyses in this paper highly depend on the choice of uncertainty sets and techniques in this paper cannot be extended easily to other uncertainty sets (e.g., noise action uncertainty sets in [Tessler, 2019]).
* It appears that the policy execution uncertainty set may not be suitable for addressing large-scale problems, as it necessitates computing the minimum over the action space. While this paper concentrates on a tabular environment, it would be more prudent to opt for a scalable uncertainty set, e.g., [Zhou et al., 2023, Natural Actor Critic for Robust Reinforcement Learning with Function Approximation].
* One important piece of literature on robust MDP is missing in the paper [Iyengar'05, Robust dynamic programming].

**Questions:**

Please refer to the ``weakness'' section for further information.

---

> ### Author Response · Authors · 2023-11-18
> **Response to Reviewer if62**
>
> **To weakness 1.**
>
> We agree with the reviewer that the concept of the probabilistic action robust MDP and the existence of its optimal policy is not novel, which was introduced by Tessler et al. (2019).
> However, Tessler et al. (2019) and other related papers did not discuss the action robust bellman equation. In Section 2, we showed that the optimal action robust policy can be solved via the iteration of the action robust Bellman optimality equation.
> This motivated us to use the action robust Bellman optimality equation to directly solve the robust problem, and then we designed two efficient algorithms based on the action robust Bellman optimality equation.
>
> Using the iteration of the proposed action robust Bellman equation to solve the robust problem can avoid the alternating update between the adversary policy and agent policy. The alternating update method is used in the approaches of Pinto et al. (2017) and Tessler et al. (2019), and is sample inefficient.
>
> Thus, we emphasize that the optimal action robust policy can be solved via the iteration of the action robust Bellman optimality equation in Section 2. and is the important idea of our proposed efficient algorithm.
>
> To reflect the above, we changed the Theorem 1 to Proposition 1 in the revised version so as to de-emphasize the results in Section 4, changed the description of the main contributions in page 2 and added the discussion about the importance of introducing action robust Bellman optimality equation in the revised paper.
>
> **To Weakness 2.**
>
> We agree with the reviewer that the regret analyses in this paper highly depend on the choice of uncertainty sets. However, in the existing literature, it is a common practice to consider the property of the specific uncertainty set when designing efficient robust RL algorithms. For example, Wang et al. (2021) consider the R-contamination probability uncertainty set, and its convergence analyses highly depend on the property of the R-contamination uncertainty set. It utilizes the property that the minimum over the probabilities set is the minimum over the entire state space.
>
> The proof techniques may not be extended easily to other uncertainty sets, but the idea of the robust algorithm designing can be potentially used in other action robust uncertainty sets. Directly using the iteration of the action robust Bellman equation and simultaneously updating adversary policy and agent policy is more sample efficient than the alternating update method in Pinto et al. (2017) and Tessler et al. (2019).
>
> **To Question 3.**
>
> This paper focuses on the tabular environment. It is true that the current theoretical guarantee on the sample complexity of the proposed algorithms is based on the tabular setting. However, the policy execution uncertainty set itself is also applicable for large-scale problems. For example, Tessler et al. (2019) introduced the probabilistic policy execution uncertainty set and perform extensive evaluation across several MuJoCo environments.
>
> Designing provably sample efficient algorithm for large-scale problems is an important future direction for us to pursue.
> From our work, we found two insights that could be useful for large-scale problems: (1) The adversary policy and the agent policy can be simultaneously updated to efficiently sample trajectories; (2) The adversary policies at each episode does not necessarily need be the minimum over the actions, an approximation of the minimum also works.
> Based on these insights, a policy gradient method could potentially be designed handle the continuous action space in the robust problem. We can use policy gradient method, such like PPO, to find an approximation of the adversary policy (the minimum over actions).
>
> We also thank the reviewer for bring the paper [Zhou et al., 2023, Natural Actor Critic for Robust Reinforcement Learning with Function Approximation] to our attention. Consider a variant of the proposed probabilistic policy execution uncertainty set is also a good idea.
> We added these discussions in the conclusion section of the revised version.
>
> **To weakness 4.**
>
> We added the discussion related to the paper [Iyengar'05, Robust dynamic programming] in the revised version.

---

> ### Comment · Reviewer_if62 · 2023-11-21
>
> Thank you for your reply to solve some of my concerns. I have increased my score.

---

> > ### Author Response · Authors · 2023-11-21
> > **Thank you for your response and for raising your score**
> >
> > Dear reviewer if62,
> >
> > We are very glad to know that our reply solves your concerns. Thank you very much for raising the score. We also greatly appreciate your valuable suggestions that make this manuscript better.

---

### Official Review · Reviewer_wBWc · 2023-11-05

**Soundness:** 2 fair
**Presentation:** 1 poor
**Contribution:** 2 fair
**Rating:** 5
**Confidence:** 4

**Summary:**

This work studies the action robust framework where a mixture policy between the maximizing agent and the minimizing adversary is effectively prescribed in the environment. Focusing on the finite horizon setting, the authors derive a Bellman recursion that solves this type of MDP and introduce ARRLC, a method that is shown to achieve minimax optimal regret. Experiments are conducted in two environments and ARRLC is compared to two baselines: (I) a non-robust algorithm called ORLC; and (ii) a robust TD algorithm. The numerical results show that ARRLC improves robustness upon ORLC while converging faster than robust TD.

**Strengths:**

This paper analyzes a setting that has essentially been addressed heuristically, to the best of my knowledge. The minimax regret and sample complexity result provide a theoretical justification for action robustness under probabilistic policy execution.

**Weaknesses:**

- The motivation for this work is unclear to me. On the one hand, action-robust MDPs have already been solved by Tessler et al. (2019) but on the other hand, this paper does not claim any improvement upon Tessler et al., nor does it numerically compare ARRLC to their solution... at least in the main body of the paper. In the appendix, there is a brief discussion on the difference between ARRLC and Tessler et al., but it is not comprehensive. Also, I do not understand why there is no convergence plot there.
- Pertaining to my previous concern, related work is described without assessing what each reference misses that this paper adds. This section should emphasize more on what previous work did and what is still missing that this study fulfills. This would not only give motivation to this work but also strengthen its significance.
- Clarity:
    - I see several grammar mistakes here and there: "A lot of robust RL methods"--> many, "the episodic RL" --> remove the; "which is deterministic" --> that is; "via the induction" --> by induction; "action ... and reward" --> actions ... and rewards; "the initial states... is deterministic... in different episode" --> are ... episodes

    - the algorithm is too far away after the text referring to it;

    - Thm. 2: "then" should follow an "if" statement

    - The uncertainty set definition in Eq. (3) is confusing: it says $\tilde{\pi}$ is a mixture between $\pi$ and $\pi'$ without specifying $\pi'$, whereas the text after states that $\pi'$ should be an argmin. Once I understood what the authors meant, I think it would have been clearer to write: $\\Pi^{\\rho}(\\pi):= \\{ \\tilde{\\pi}: \\forall s, \\forall h, \\exists \\pi_{h}'(\\cdot|s) \\in\\Delta_{\\mathcal{A}} \\text{ such that } $ $\\tilde{\\pi_{h}}$ $(\\cdot|s)=$ $(1-\\rho)\\pi_{h}(\\cdot|s)+ \\rho \\pi_{h}'(\\cdot |s) \\},$ or even simpler, $\\Pi^{\\rho}(\\pi):=  (1-\\rho)\\pi + \\rho $ $(\\Delta_{\\mathcal{A}})^{|\\mathcal{S}|\\times H}$

- It is not clear to me how this work improves upon Tessler et al.. It does give regret bounds and sample complexity, but how is the algorithmic approach essentially different?

**Questions:**

- Why did the authors not compare ARRLC to the approaches of Pinto et al. (2017) or Tessler et al. (2019) in the experiments section, but chose a comparison to a non-robust approach ORLC instead?
- I simply remark that $\\Pi^{\\rho}\\subset\\Pi^{D, \\rho}$, is that correct? Also, what does the relationship between those two types of uncertainty sets bring to the analysis?
- As far as I remember, Tessler et al. (2019) proved the existence of an optimal policy for the action robust setting under probabilistic execution. Although their proof does not rely on Bellman recursion but a Markov game formulation, they show equivalence between this setting and the robust MDP setting, itself being solvable through Bellman recursion [1]. Could the authors please clarify?

[1] Iyengar, Garud N. "Robust dynamic programming." Mathematics of Operations Research 30.2 (2005): 257-280.

---

> ### Author Response · Authors · 2023-11-18
> **Response to Reviewer wBWc (1)**
>
> **To weakness 1.**
>
> Although the probabilistic action robust MDP was discussed in Tessler et al. (2019), no theoretical guarantee on sample complexity or regret was provided in Tessler et al. (2019).
> The methods in Tessler et al. (2019), i.e. PR-PI and soft PR-PI, iteratively repeat two stages: (i) given a fixed adversary policy, it calculates the optimal counter policy, and (ii) update the adversary policy against the updated counter policy. Stage (i) may be performed by any MDP solver. Tessler et al. (2019) shows the convergence of this iterative approach. However, this form of iteratively updating requires repeating stage (1), i.e. repeatedly solving MDP to find the optimal policy. It is sample inefficient and computational inefficient.
>
> In this paper, we fill this gap by designing a provably efficient action robust RL algorithm. Instead of modeling the problem as a Markov Game, we solve the minimax problem by the iteration of the proposed action robust Bellman equation. In our method, the adversary policy and agent policy are updated simultaneously. We update the adversary policy in each episode even if the agent policy is not optimal under the current adversary policy. At the same time, we update the agent policy in each episode even if the current adversary policy does not minimize the rewards. We prove that the adversary policy and agent policy gradually converge to the solution together. By this, ARRLC directly learn the robust policy. We further prove the efficiency and sample optimality of ARRLC.
>
> **To weakness 2.**
>
> We updated the discussion on the related works in the revised version (marked in red color).
> The main update are: (1) the previous works on action robust RL fail to design efficient algorithm with theoretical guarantee on sample complexity or regret. We develop a minimax sample efficient algorithm and fill in this gap. (2) the works on sample complexity guarantees for episodic tabular RL can not directly use in action robust MDP with small technical changes.  We develop new adversarial trajectory sampling and action robust value iteration method in the proposed algorithm, and new techniques to bound the sum of variance so that our algorithm suits for action robust MDPs. (3) the efficient multi-agent RL algorithms, such as (Liu et al., 2021; Jin et al., 2021), can be used to solve the action robust optimal policy but are not minimax optimal. They are a factor of $A$ or $H^2$ above the minimax lower bound. Our algorithm ARRLC is minimax optimal.
>
> **To weakness 3.**
>
> We fixed typos in the revised version and changed the position of Algorithm 1. We agree with the reviewer that the original uncertainty set definition in Eq. (3) may by confusing. We have rewritten it in a form as the reviewer suggested.
>
> **To weakness 4.**
>
> Regarding the algorithmic approach, Tessler et al. (2019) requires repeatedly solving the optimal policy given a fixed adversary policy, which is sample inefficient and computationally inefficient. Our algorithm solves the robust problem by the iteration of the proposed action robust Bellman equation. The proposed algorithm directly learns the robust policy and simultaneously updates the adversary policy and the agent policy without fixing any of them.

---

> ### Author Response · Authors · 2023-11-18
> **Response to Reviewer wBWc (2)**
>
> **To Question 1.**
>
> Our experimental results consists of three parts.
> (1) We compared our proposed method ARRLC with a non-robust algorithm ORLC and show the robustness of our algorithm.
> (2) We compared ARRLC with robust TD and demonstrates the efficiency of our ARRLC algorithm. ARRLC converges much faster than robust TD algorithm.
> (3) In the appendix, we compared our algorithm with the algorithms in Pinto et al. (2017) and Tessler et al. (2019) under different $\rho$ and $p$. We used a black-box MDP solver (Q-learning) to solve the optimal agent policy when the adversarial policy is given. We trained our method in 2000 episodes and others in 30000 episodes. Compared with our algorithm, the approaches of Pinto et al. (2017) and Tessler et al. (2019) are sample inefficient .
>
> Due to the page limits, we put some experimental results of (3) into Appendix. We agree with the reviewer that providing the numerical comparison with the approaches of Pinto et al. (2017) and Tessler et al. (2019) in the main body of the paper can provide more insights. Hence, in the revised paper, we reorganized the experiment section. We moved the the numerical comparison with the approaches of Pinto et al. (2017) and Tessler et al. (2019) into the main body of the paper, and moved other experimental results into Appendix due to the page limit.
>
> **To Question 2.**
>
> We agree with the reviewer. The relationship between those two types of uncertainty sets does not impact the analysis. We deleted this discussion in the revised version.
>
> **To Question 3.**
>
> We agree with the reviewer that Tessler et al. (2019) proved the existence of an optimal policy for the action robust setting under probabilistic execution and showed the equivalence between this setting and one special robust MDP setting.
>
> However the equivalent robust MDP setting in Tessler et al. (2019) is different with the setting in Iyengar et al. (2005). The equivalent robust MDP setting in Tessler et al. (2019) includes both the reward uncertainty and the transition probabilities uncertainty, while the Robust MDP setting in Iyengar et al. (2005) only considers the uncertainty in transition probabilities. Thus, the robust Bellman equations in two papers are different and the results in Iyengar et al. (2005) can not directly apply in action robust setting.

---

> ### Author Response · Authors · 2023-11-21
> **A kind remind**
>
> Dear reviewer wBWc,
>
> Thank you for dedicating your time to reviewing our manuscript. Your insights have been invaluable, and we've crafted a detailed response to address the points you raised and polish the paper accordingly. We're confident that our responses and modifications have effectively addressed your concerns. With these changes in mind, we hope you might reconsider and adjust your evaluation of our work.
>
> Should there be any remaining issues, we're more than ready to provide further clarifications. As the rebuttal phase is nearing its deadline, we're looking forward to engaging in a timely discussion. Thank you again for your expertise and guidance!

---

### Author Response · Authors · 2023-11-18
**Changes in the revised version.**

We thank the reviewers for their valuable feedback, suggestions, and time invested. Here, we introduce the changes in the revised version, which are marked in red color.

1. We added more discussions about the motivation of our work. We compared our work with [Pinto et al. (2017), Tessler et al. (2019)] not only on the contribution but also on the algorithmic approach differences.

2. We updated the discussion on the related works in the revised version.

3. We changed the Theorem 1 to Proposition 1 in the revised version so as to de-emphasize the results in Section 4, changed the description of the main contributions in page 2 and added the discussion about the importance of introducing action robust Bellman optimality equation in the revised paper.

4. In the revised paper, we reorganized the experiment section. We moved the the numerical comparison with the approaches of Pinto et al. (2017) and Tessler et al. (2019) into the main body of the paper, and moved other experimental results into Appendix due to the page limit.

5. We added the discussions about the scalability of our algorithm in the conclusion section of the revised version.

---

### Meta-Review · Area_Chair_rSjs · 2023-12-09

**Metareview:**

This paper studies the action-robust finite-horizon MDP problem where a mixture policy between the maximizing agent and the minimizing adversary is present. The authors introduced ARRLC, a method that achieves minimax optimal regret with experiments on two environments that outperforms two standard baselines: non-robust ORLC and robust TD in both improved robustness and faster convergence.

Paper present interesting ideas. However, reviewers concern about limited motivations/novelty of this work, significance of contributions, soundness of regret analysis with the specific choices of uncertainty set, additional baselines in robust MDPs (for example the 2005 work by Iyengar) etc. Clarity of the paper and literature review should also be improved with a revision. Overall, paper is on the borderline of acceptance/rejection but is worth another revision before resubmission.

**Justification For Why Not Higher Score:**

There are still significant issues raised by reviewers (mentioned above) that needs to be addressed before it can be reconsidered for acceptance.

**Justification For Why Not Lower Score:**

N/A

---

### Decision · Program_Chairs · 2024-01-16

Reject